# AUGMENTING PHYSICAL MODELS WITH DEEP NETWORKS FOR COMPLEX DYNAMICS FORECASTING

[*]**Yuan Yin**[1]      [*]**Vincent Le Guen**[2,3]      [*]**Jérémie Dona**[1]      [*]**Emmanuel de Bézenac**[1]
[*]**Ibrahim Ayed**[1,4]      **Nicolas Thome**[2]      **Patrick Gallinari**[1,5]

[1] Sorbonne Université, CNRS, LIP6, Paris, France
[2] Conservatoire National des Arts et Métiers, CEDRIC, Paris, France
[3] EDF R&D, Chatou, France
[4] Theresis Lab, Thales
[5] Criteo AI Lab, Paris, France

## ABSTRACT

Forecasting complex dynamical phenomena in settings where only partial knowledge of their dynamics is available is a prevalent problem across various scientific fields. While purely data-driven approaches are arguably insufficient in this context, standard physical modeling based approaches tend to be over-simplistic, inducing non-negligible errors. In this work, we introduce the APHYNITY framework, a principled approach for augmenting *incomplete* physical dynamics described by differential equations with deep data-driven models. It consists in decomposing the dynamics into two components: a physical component accounting for the dynamics for which we have some prior knowledge, and a data-driven component accounting for errors of the physical model. The learning problem is carefully formulated such that the physical model explains as much of the data as possible, while the data-driven component only describes information that cannot be captured by the physical model, no more, no less. This not only provides the existence and uniqueness for this decomposition, but also ensures interpretability and benefits generalization. Experiments made on three important use cases, each representative of a different family of phenomena, i.e. reaction-diffusion equations, wave equations and the non-linear damped pendulum, show that APHYNITY can efficiently leverage approximate physical models to accurately forecast the evolution of the system and correctly identify relevant physical parameters.

## 1   INTRODUCTION

Modeling and forecasting complex dynamical systems is a major challenge in domains such as environment and climate (Rolnick et al., 2019), health science (Choi et al., 2016), and in many industrial  applications (Toubeau et al., 2018). Model Based (MB) approaches typically rely on partial or ordinary differential equations (PDE/ODE) and stem from a deep understanding of the underlying physical phenomena. Machine learning (ML) and deep learning methods are more prior agnostic yet have become state-of-the-art for several spatio-temporal prediction tasks (Shi et al., 2015; Wang et al., 2018; Oreshkin et al., 2020; Donà et al., 2020), and connections have been drawn between deep architectures and numerical ODE solvers, e.g. neural ODEs (Chen et al., 2018; Ayed et al., 2019b). However, modeling complex physical dynamics is still beyond the scope of pure ML methods, which often cannot properly extrapolate to new conditions as MB approaches do.

Combining the MB and ML paradigms is an emerging trend to develop the interplay between the two paradigms. For example, Brunton et al. (2016); Long et al. (2018b) learn the explicit form of PDEs directly from data, Raissi et al. (2019); Sirignano & Spiliopoulos (2018) use NNs as implicit methods for solving PDEs, Seo et al. (2020) learn spatial differences with a graph network, Ummenhofer et al. (2020) introduce continuous convolutions for fluid simulations, de Bézenac et al. (2018) learn the

---

[*]Equal contribution, authors sorted by reverse alphabetical order.

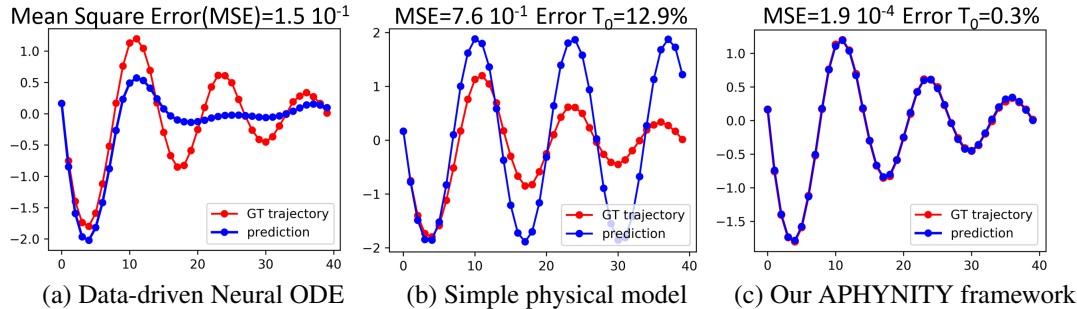

Figure 1: Predicted dynamics for the damped pendulum vs. ground truth (GT) trajectories $\mathrm{d}^2\theta/\mathrm{d}t^2 + \omega_0^2 \sin\theta + \alpha \mathrm{d}\theta/\mathrm{d}t = 0$. We show that in (a) the data-driven approach (Chen et al., 2018) fails to properly learn the dynamics due to the lack of training data, while in (b) an ideal pendulum cannot take friction into account. The proposed APHYNITY shown in (c) augments the over-simplified physical model in (b) with a data-driven component. APHYNITY improves both forecasting (MSE) and parameter identification (Error $T_0$) compared to (b).

velocity field of an advection-diffusion system, Greydanus et al. (2019); Chen et al. (2020) enforce conservation laws in the network architecture or in the loss function.

The large majority of aforementioned MB/ML hybrid approaches assume that the physical model adequately describes the observed dynamics. This assumption is, however, commonly violated in practice. This may be due to various factors, e.g. idealized assumptions and difficulty to explain processes from first principles (Gentine et al., 2018), computational constraints prescribing a fine grain modeling of the system (Ayed et al., 2019a), unknown external factors, forces and sources which are present (Large & Yeager, 2004). In this paper, we aim at leveraging prior dynamical ODE/PDE knowledge in situations where this physical model is incomplete, i.e. unable to represent the whole complexity of observed data. To handle this case, we introduce a principled learning framework to Augment incomplete PHYsical models for ideNtIfying and forecasTing complex dYnamics (APHYNITY). The rationale of APHYNITY, illustrated in Figure 1 on the pendulum problem, is to *augment* the physical model when—and only when—it falls short.

Designing a general method for combining MB and ML approaches is still a widely open problem, and a clear problem formulation for the latter is lacking (Reichstein et al., 2019). Our contributions towards these goals are the following:

- We introduce a simple yet principled framework for combining both approaches. We decompose the data into a physical and a data-driven term such that the data-driven component only models information that cannot be captured by the physical model. We provide existence and uniqueness guarantees (Section 3.1) for the decomposition given mild conditions, and show that this formulation ensures interpretability and benefits generalization.

- We propose a trajectory-based training formulation (Section 3.2) along with an adaptive optimization scheme (Section 3.3) enabling end-to-end learning for both physical and deep learning components. This allows APHYNITY to *automatically* adjust the complexity of the neural network to different approximation levels of the physical model, paving the way to flexible learned hybrid models.

- We demonstrate the generality of the approach on three use cases (reaction-diffusion, wave equations and the pendulum) representative of different PDE families (parabolic, hyperbolic), having a wide spectrum of application domains, e.g. acoustics, electromagnetism, chemistry, biology, physics (Section 4). We show that APHYNITY is able to achieve performances close to complete physical models by augmenting incomplete ones, both in terms of forecasting accuracy and physical parameter identification. Moreover, APHYNITY can also be successfully extended to the partially observable setting (see discussion in Section 5).

## 2 RELATED WORK

**Correction in data assimilation** Prediction under approximate physical models has been tackled by traditional statistical calibration techniques, which often rely on Bayesian methods (Pernot & Cailliez, 2017). Data assimilation techniques, e.g. the Kalman filter (Kalman, 1960; Becker et al., 2019), 4D-var (Courtier et al., 1994), prediction errors are modeled probabilistically and a correction using observed data is applied after each prediction step. Similar residual correction procedures are commonly used in robotics and optimal control (Chen, 2004; Li et al., 2014). However, these sequential (two-stage) procedures prevent the cooperation between prediction and correction. Besides, in model-based reinforcement learning, model deficiencies are typically handled by considering only short-term rollouts (Janner et al., 2019) or by model predictive control (Nagabandi et al., 2018). The originality of APHYNITY is to leverage model-based prior knowledge by augmenting it with neurally parametrized dynamics. It does so while ensuring optimal cooperation between the prior model and the augmentation.

**Augmented physical models** Combining physical models with machine learning (*gray-box or hybrid* modeling) was first explored from the 1990's: Psichogios & Ungar (1992); Thompson & Kramer (1994); Rico-Martinez et al. (1994) use neural networks to predict the unknown parameters of physical models. The challenge of proper MB/ML cooperation was already raised as a limitation of gray-box approaches but not addressed. Moreover these methods were evaluated on specific applications with a residual targeted to the form of the equation. In the last few years, there has been a renewed interest in deep hybrid models bridging data assimilation techniques and machine learning to identify complex PDE parameters using cautiously constrained forward model (Long et al., 2018b; de Bézenac et al., 2018), as discussed in introduction. Recently, some approaches have specifically targetted the MB/ML cooperation. HybridNet (Long et al., 2018a) and PhICNet (Saha et al., 2020) both use data-driven networks to learn additive perturbations or source terms to a given PDE. The former considers the favorable context where the perturbations can be accessed, and the latter the special case of additive noise on the input. Wang et al. (2019); Mehta et al. (2020) propose several empirical fusion strategies with deep neural networks but lack theoretical groundings. PhyDNet (Le Guen & Thome, 2020) tackles augmentation in partially-observed settings, but with specific recurrent architectures dedicated to video prediction. Crucially, all the aforementioned approaches do not address the issues of uniqueness of the decomposition or of proper cooperation for correct parameter identification. Besides, we found experimentally that this vanilla cooperation is inferior to the APHYNITY learning scheme in terms of forecasting and parameter identification performances (see experiments in Section 4.2).

## 3 THE APHYNITY MODEL

In the following, we study dynamics driven by an equation of the form:

$$\frac{\mathrm{d}X_t}{\mathrm{d}t} = F(X_t) \tag{1}$$

defined over a finite time interval $[0, T]$, where the state $X$ is either vector-valued, i.e. we have $X_t \in \mathbb{R}^d$ for every $t$, (pendulum equations in Section 4), or $X_t$ is a $d$-dimensional vector field over a spatial domain $\Omega \subset \mathbb{R}^k$, with $k \in \{2, 3\}$, i.e. $X_t(x) \in \mathbb{R}^d$ for every $(t, x) \in [0, T] \times \Omega$ (reaction-diffusion and wave equations in Section 4). We suppose that we have access to a set of observed trajectories $\mathcal{D} = \{X. : [0, T] \to \mathcal{A} \mid \forall t \in [0, T], \mathrm{d}X_t/\mathrm{d}t = F(X_t)\}$, where $\mathcal{A}$ is the set of $X$ values (either $\mathbb{R}^d$ or vector field). In our case, the unknown $F$ has $\mathcal{A}$ as domain and we only assume that $F \in \mathcal{F}$, with $(\mathcal{F}, \|\cdot\|)$ a normed vector space.

### 3.1 DECOMPOSING DYNAMICS INTO PHYSICAL AND AUGMENTED TERMS

As introduced in Section 1, we consider the common situation where incomplete information is available on the dynamics, under the form of a family of ODEs or PDEs characterized by their temporal evolution $F_p \in \mathcal{F}_p \subset \mathcal{F}$. The APHYNITY framework leverages the knowledge of $\mathcal{F}_p$ while mitigating the approximations induced by this simplified model through the combination of physical and data-driven components. $\mathcal{F}$ being a vector space, we can write:

$$F = F_p + F_a$$

where $F_p \in \mathcal{F}_p$ encodes the incomplete physical knowledge and $F_a \in \mathcal{F}$ is the data-driven augmentation term complementing $F_p$. The incomplete physical prior is supposed to belong to a known family, but the physical parameters (e.g. propagation speed for the wave equation) are unknown and need to be estimated from data. Both $F_p$ and $F_a$ parameters are estimated by fitting the trajectories from $\mathcal{D}$.

The decomposition $F = F_p + F_a$ is in general not unique. For example, all the dynamics could be captured by the $F_a$ component. This decomposition is thus ill-defined, which hampers the interpretability and the extrapolation abilities of the model. In other words, one wants the estimated parameters of $F_p$ to be as close as possible to the true parameter values of the physical model and $F_a$ to play only a complementary role w.r.t $F_p$, so *as to model only the information that cannot be captured by the physical prior*. For example, when $F \in \mathcal{F}_p$, the data can be fully described by the physical model, and in this case it is sensible to desire $F_a$ to be nullified; this is of central importance in a setting where one wishes to identify physical quantities, and for the model to generalize and extrapolate to new conditions. In a more general setting where the physical model is incomplete, the action of $F_a$ on the dynamics, as measured through its norm, should be as small as possible.

This general idea is embedded in the following optimization problem:

$$\min_{F_p \in \mathcal{F}_p, F_a \in \mathcal{F}} \quad \|F_a\| \quad \text{subject to} \quad \forall X \in \mathcal{D}, \forall t, \frac{\mathrm{d}X_t}{\mathrm{d}t} = (F_p + F_a)(X_t) \tag{2}$$

The originality of APHYNITY is to leverage model-based prior knowledge by augmenting it with neurally parametrized dynamics. It does so while ensuring optimal cooperation between the prior model and the augmentation.

A first key question is whether the minimum in Eq. (2) is indeed well-defined, in other words whether there exists indeed a decomposition with a minimal norm $F_a$. The answer actually depends on the geometry of $\mathcal{F}_p$, and is formulated in the following proposition proven in Appendix B:

**Proposition 1** (Existence of a minimizing pair). *If $\mathcal{F}_p$ is a proximinal set[1], there exists a decomposition minimizing Eq. (2).*

Proximinality is a mild condition which, as shown through the proof of the proposition, cannot be weakened. It is a property verified by any boundedly compact set. In particular, it is true for closed subsets of finite dimensional spaces. However, if only existence is guaranteed, while forecasts would be expected to be accurate, non-uniqueness of the decomposition would hamper the interpretability of $F_p$ and this would mean that the identified physical parameters are not uniquely determined.

It is then natural to ask under which conditions solving problem Eq. (2) leads to a unique decomposition into a physical and a data-driven component. The following result provides guarantees on the existence and uniqueness of the decomposition under mild conditions. The proof is given in Appendix B:

**Proposition 2** (Uniqueness of the minimizing pair). *If $\mathcal{F}_p$ is a Chebyshev set[1], Eq. (2) admits a unique minimizer. The $F_p$ in this minimizer pair is the metric projection of the unknown $F$ onto $\mathcal{F}_p$.*

The Chebyshev assumption condition is strictly stronger than proximinality but is still quite mild and necessary. Indeed, in practice, many sets of interest are Chebyshev, including all closed convex spaces in strict normed spaces and, if $\mathcal{F} = L^2$, $\mathcal{F}_p$ can be any closed convex set, including all finite dimensional subspaces. In particular, all examples considered in the experiments are Chebyshev sets.

Propositions 1 and 2 provide, under mild conditions, the theoretical guarantees for the APHYNITY formulation to infer the correct MB/ML decomposition, thus enabling both recovering the proper physical parameters and accurate forecasting.

## 3.2 Solving APHYNITY with deep neural networks

In the following, both terms of the decomposition are parametrized and are denoted as $F_p^{\theta_p}$ and $F_p^{\theta_a}$. Solving APHYNITY then consists in estimating the parameters $\theta_p$ and $\theta_a$. $\theta_p$ are the physical parameters and are typically low-dimensional, e.g. 2 or 3 in our experiments for the considered physical models. For $F_a$, we need sufficiently expressive models able to optimize over all $\mathcal{F}$: we

---

[1]A proximinal set is one from which every point of the space has at least one nearest point. A Chebyshev set is one from which every point of the space has a unique nearest point. More details in Appendix A.

thus use deep neural networks, which have shown promising performances for the approximation of differential equations (Raissi et al., 2019; Ayed et al., 2019b).

When learning the parameters of $F_p^{\theta_p}$ and $F_a^{\theta_a}$, we have access to a finite dataset of trajectories discretized with a given temporal resolution $\Delta t$: $\mathcal{D}_{\text{train}} = \{(X_{k\Delta t}^{(i)})_{0 \leq k \leq \lfloor T/\Delta t \rfloor}\}_{1 \leq i \leq N}$. Solving Eq. (2) requires estimating the state derivative $\mathrm{d}X_t/\mathrm{d}t$ appearing in the constraint term. One solution is to approximate this derivative using e.g. finite differences as in (Brunton et al., 2016; Greydanus et al., 2019; Cranmer et al., 2020). This numerical scheme requires high space and time resolutions in the observation space in order to get reliable gradient estimates. Furthermore it is often unstable, leading to explosive numerical errors as discussed in Appendix D. We propose instead to solve Eq. (2) using an integral trajectory-based approach: we compute $\widetilde{X}_{k\Delta t, X_0}^i$ from an initial state $X_0^{(i)}$ using the current $F_p^{\theta_p} + F_a^{\theta_a}$ dynamics, then enforce the constraint $\widetilde{X}_{k\Delta t, X_0}^i = X_{k\Delta t}^i$. This leads to our final objective function on $(\theta_p, \theta_a)$:

$$\min_{\theta_p, \theta_a} \quad \left\| F_a^{\theta_a} \right\| \quad \text{subject to} \quad \forall i, \forall k, \widetilde{X}_{k\Delta t}^{(i)} = X_{k\Delta t}^{(i)} \tag{3}$$

where $\widetilde{X}_{k\Delta t}^{(i)}$ is the approximate solution of the integral $\int_{X_0^{(i)}}^{X_0^{(i)} + k\Delta t} (F_p^{\theta_p} + F_a^{\theta_a})(X_s)\, \mathrm{d}X_s$ obtained by a differentiable ODE solver.

In our setting, where we consider situations for which $F_p^{\theta_p}$ only partially describes the physical phenomenon, this coupled MB + ML formulation leads to different parameter estimates than using the MB formulation alone, as analyzed more thoroughly in Appendix C. Interestingly, our experiments show that using this formulation also leads to a better identification of the physical parameters $\theta_p$ than when fitting the simplified physical model $F_p^{\theta_p}$ alone (Section 4). With only an incomplete knowledge on the physics, $\theta_p$ estimator will be biased by the additional dynamics which needs to be fitted in the data. Appendix F also confirms that the integral formulation gives better forecasting results and a more stable behavior than supervising over finite difference approximations of the derivatives.

### 3.3 ADAPTIVELY CONSTRAINED OPTIMIZATION

The formulation in Eq. (3) involves constraints which are difficult to enforce exactly in practice. We considered a variant of the method of multipliers (Bertsekas, 1996) which uses a sequence of Lagrangian relaxations $\mathcal{L}_{\lambda_j}(\theta_p, \theta_a)$:

$$\mathcal{L}_{\lambda_j}(\theta_p, \theta_a) = \left\| F_a^{\theta_a} \right\| + \lambda_j \cdot \mathcal{L}_{traj}(\theta_p, \theta_a) \tag{4}$$

where $\mathcal{L}_{traj}(\theta_p, \theta_a) = \sum_{i=1}^{N} \sum_{h=1}^{T/\Delta t} \| X_{h\Delta t}^{(i)} - \widetilde{X}_{h\Delta t}^{(i)} \|$.

This method needs an increasing sequence $(\lambda_j)_j$ such that the successive minima of $\mathcal{L}_{\lambda_j}$ converge to a solution (at least a local one) of the constrained problem Eq. (3). We select $(\lambda_j)_j$ by using an iterative strategy: starting from a value $\lambda_0$, we iterate, minimizing $\mathcal{L}_{\lambda_j}$ by gradient descent[2], then update $\lambda_j$ with: $\lambda_{j+1} = \lambda_j + \tau_2 \mathcal{L}_{traj}(\theta_{j+1})$, where $\tau_2$ is a chosen hyper-parameter and $\theta = (\theta_p, \theta_a)$. This procedure is summarized in Algorithm 1. This adaptive iterative procedure allows us to obtain stable and robust results, in a reproducible fashion, as shown in the experiments.

---

**Algorithm 1:** APHYNITY

Initialization: $\lambda_0 \geq 0, \tau_1 > 0, \tau_2 > 0$;
**for** $epoch = 1 : N_{epochs}$ **do**
  **for** $iter$ in $1 : N_{iter}$ **do**
    **for** $batch$ in $1 : B$ **do**
      $\theta_{j+1} = \theta_j - $
        $\tau_1 \nabla [\lambda_j \mathcal{L}_{traj}(\theta_j) + \|F_a\|]$
  $\lambda_{j+1} = \lambda_j + \tau_2 \mathcal{L}_{traj}(\theta_{j+1})$

---

## 4 EXPERIMENTAL VALIDATION

We validate our approach on 3 classes of challenging physical dynamics: reaction-diffusion, wave propagation, and the damped pendulum, representative of various application domains such as chemistry, biology or ecology (for reaction-diffusion) and earth physic, acoustic, electromagnetism or

---

[2]Convergence to a local minimum isn't necessary, a few steps are often sufficient for a successful optimization.

even neuro-biology (for waves equations). The two first dynamics are described by PDEs and thus in practice should be learned from very high-dimensional vectors, discretized from the original compact domain. This makes the learning much more difficult than from the one-dimensional pendulum case. For each problem, we investigate the cooperation between physical models of increasing complexity encoding incomplete knowledge of the dynamics (denoted *Incomplete physics* in the following) and data-driven models. We show the relevance of APHYNITY (denoted *APHYNITY models*) both in terms of forecasting accuracy and physical parameter identification.

## 4.1 EXPERIMENTAL SETTING

We describe the three families of equations studied in the experiments. In all experiments, $\mathcal{F} = \mathcal{L}^2(\mathcal{A})$ where $\mathcal{A}$ is the set of all admissible states for each problem, and the $\mathcal{L}^2$ norm is computed on $\mathcal{D}_{train}$ by: $\|F\|^2 \approx \sum_{i,k} \|F(X_{k\Delta t}^{(i)})\|^2$. All considered sets of physical functionals $\mathcal{F}_p$ are closed and convex in $\mathcal{F}$ and thus are Chebyshev. In order to enable the evaluation on both prediction and parameter identification, all our experiments are conducted on simulated datasets with known model parameters. Each dataset has been simulated using an appropriate high-precision integration scheme for the corresponding equation. All solver-based models take the first state $X_0$ as input and predict the remaining time-steps by integrating $F$ through the same differentiable generic and common ODE solver (4th order Runge-Kutta)[3]. Implementation details and architectures are given in Appendix E.

**Reaction-diffusion equations** We consider a 2D FitzHugh-Nagumo type model (Klaasen & Troy, 1984). The system is driven by the PDE $\frac{\partial u}{\partial t} = a\Delta u + R_u(u, v; k)$, $\frac{\partial v}{\partial t} = b\Delta v + R_v(u, v)$ where $a$ and $b$ are respectively the diffusion coefficients of $u$ and $v$, $\Delta$ is the Laplace operator. The local reaction terms are $R_u(u, v; k) = u - u^3 - k - v$, $R_v(u, v) = u - v$. The state is $X = (u, v)$ and is defined over a compact rectangular domain $\Omega$ with periodic boundary conditions. The considered physical models are: • *Param PDE $(a, b)$*, with unknown $(a, b)$ diffusion terms and without reaction terms: $\mathcal{F}_p = \{F_p^{a,b} : (u, v) \mapsto (a\Delta u, b\Delta v) \mid a \geq a_{\min} > 0, b \geq b_{\min} > 0\}$; • *Param PDE $(a, b, k)$*, the full PDE with unknown parameters: $\mathcal{F}_p = \{F_p^{a,b,k} : (u, v) \mapsto (a\Delta u + R_u(u, v; k), b\Delta v + R_v(u, v) \mid a \geq a_{\min} > 0, b \geq b_{\min} > 0, k \geq k_{\min} > 0\}$.

**Damped wave equations** We investigate the damped-wave PDE: $\frac{\partial^2 w}{\partial t^2} - c^2 \Delta w + k\frac{\partial w}{\partial t} = 0$ where $k$ is the damping coefficient. The state is $X = (w, \frac{\partial w}{\partial t})$ and we consider a compact spatial domain $\Omega$ with Neumann homogeneous boundary conditions. Note that this damping differs from the pendulum, as its effect is global. Our physical models are: • *Param PDE $(c)$*, without damping term: $\mathcal{F}_p = \{F_p^c : (u, v) \mapsto (v, c^2\Delta u) \mid c \in [\epsilon, +\infty) \text{ with } \epsilon > 0\}$; • *Param PDE $(c, k)$*: $\mathcal{F}_p = \{F_p^{c,k} : (u, v) \mapsto (v, c^2\Delta u - kv) \mid c, k \in [\epsilon, +\infty) \text{ with } \epsilon > 0\}$.

**Damped pendulum** The evolution follows the ODE $\mathrm{d}^2\theta/\mathrm{d}t^2 + \omega_0^2 \sin\theta + \alpha\mathrm{d}\theta/\mathrm{d}t = 0$, where $\theta(t)$ is the angle, $\omega_0$ the proper pulsation ($T_0$ the period) and $\alpha$ the damping coefficient. With state $X = (\theta, \mathrm{d}\theta/\mathrm{d}t)$, the ODE is $F_p^{\omega_0,\alpha} : X \mapsto (\mathrm{d}\theta/\mathrm{d}t, -\omega_0^2\sin\theta - \alpha\mathrm{d}\theta/\mathrm{d}t)$. Our physical models are: • *Hamiltonian* (Greydanus et al., 2019), a conservative approximation, with $\mathcal{F}_p = \{F_p^{\mathcal{H}} : (u, v) \mapsto (\partial_y\mathcal{H}(u, v), -\partial_x\mathcal{H}(u, v)) \mid \mathcal{H} \in H^1(\mathbb{R}^2)\}$, $H^1(\mathbb{R}^2)$ is the first order Sobolev space. • *Param ODE $(\omega_0)$*, the frictionless pendulum: $\mathcal{F}_p = \{F_p^{\omega_0,\alpha=0} \mid \omega_0 \in [\epsilon, +\infty) \text{ with } \epsilon > 0\}$ • *Param ODE $(\omega_0, \alpha)$*, the full pendulum equation: $\mathcal{F}_p = \{F_p^{\omega_0,\alpha} \mid \omega_0, \alpha \in [\epsilon, +\infty) \text{ with } \epsilon > 0\}$.

**Baselines** As purely data-driven baselines, we use Neural ODE (Chen et al., 2018) for the three problems and PredRNN++ (Wang et al., 2018, for reaction-diffusion only) which are competitive models for datasets generated by differential equations and for spatio-temporal data. As MB/ML methods, in the ablations studies (see Appendix F), we compare for all problems, to the vanilla MB/ML cooperation scheme found in (Wang et al., 2019; Mehta et al., 2020). We also show results for *True PDE/ODE*, which corresponds to the equation for data simulation (which do not lead to zero error due to the difference between simulation and training integration schemes). For the pendulum, we compare to Hamiltonian neural networks (Greydanus et al., 2019; Toth et al., 2020) and to the the deep Galerkin method (DGM, Sirignano & Spiliopoulos, 2018). See additional details in Appendix E.

Table 1: Forecasting and identification results on the (a) reaction-diffusion, (b) wave equation, and (c) damped pendulum datasets. We set for (a) $a = 1 \times 10^{-3}, b = 5 \times 10^{-3}, k = 5 \times 10^{-3}$, for (b) $c = 330, k = 50$ and for (c) $T_0 = 6, \alpha = 0.2$ as true parameters. $\log$ MSEs are computed respectively over 25, 25, and 40 predicted time-steps. %Err param. averages the results when several physical parameters are present. For each level of incorporated physical knowledge, equivalent best results according to a Student t-test are shown in bold. n/a corresponds to non-applicable cases.

| Dataset | | Method | log MSE | %Err param. | $\|F_a\|^2$ |
|---|---|---|---|---|---|
| **(a)** **Reaction-** **diffusion** | Data- driven | Neural ODE | -3.76±0.02 | n/a | n/a |
| | | PredRNN++ | -4.60±0.01 | n/a | n/a |
| | Incomplete physics | Param PDE $(a, b)$ | -1.26±0.02 | 67.6 | n/a |
| | | APHYNITY Param PDE $(a, b)$ | **-5.10±0.21** | **2.3** | 67 |
| | Complete physics | Param PDE $(a, b, k)$ | **-9.34±0.20** | 0.17 | n/a |
| | | APHYNITY Param PDE $(a, b, k)$ | **-9.35±0.02** | **0.096** | 1.5e-6 |
| | | True PDE | -8.81±0.05 | n/a | n/a |
| | | APHYNITY True PDE | **-9.17±0.02** | n/a | 1.4e-7 |
| **(b)** **Wave** **equation** | Data-driven | Neural ODE | -2.51±0.29 | n/a | n/a |
| | Incomplete physics | Param PDE $(c)$ | 0.51±0.07 | 10.4 | n/a |
| | | APHYNITY Param PDE $(c)$ | **-4.64±0.25** | **0.31** | 71. |
| | Complete physics | Param PDE $(c, k)$ | -4.68±0.55 | 1.38 | n/a |
| | | APHYNITY Param PDE $(c, k)$ | **-6.09±0.28** | **0.70** | 4.54 |
| | | True PDE | -4.66±0.30 | n/a | n/a |
| | | APHYNITY True PDE | **-5.24±0.45** | n/a | 0.14 |
| **(c)** **Damped** **pendulum** | Data-driven | Neural ODE | -2.84±0.70 | n/a | n/a |
| | Incomplete physics | Hamiltonian | -0.35±0.10 | n/a | n/a |
| | | APHYNITY Hamiltonian | **-3.97±1.20** | n/a | 623 |
| | | Param ODE $(\omega_0)$ | -0.14±0.10 | 13.2 | n/a |
| | | Deep Galerkin Method $(\omega_0)$ | -3.10±0.40 | 22.1 | n/a |
| | | APHYNITY Param ODE $(\omega_0)$ | **-7.86±0.60** | **4.0** | 132 |
| | Complete physics | Param ODE $(\omega_0, \alpha)$ | **-8.28±0.40** | **0.45** | n/a |
| | | Deep Galerkin Method $(\omega_0, \alpha)$ | -3.14±0.40 | 7.1 | n/a |
| | | APHYNITY Param ODE $(\omega_0, \alpha)$ | **-8.31±0.30** | **0.39** | 8.5 |
| | | True ODE | **-8.58±0.20** | n/a | n/a |
| | | APHYNITY True ODE | **-8.44±0.20** | n/a | 2.3 |

## 4.2 RESULTS

We analyze and discuss below the results obtained for the three kind of dynamics. We successively examine different evaluation or quality criteria. The conclusions are consistent for the three problems, which allows us to highlight clear trends for all of them.

**Forecasting accuracy** The data-driven models do not perform well compared to *True PDE/ODE* (all values are test errors expressed as $\log$ MSE): -4.6 for PredRNN++ vs. -9.17 for reaction-diffusion, -2.51 vs. -5.24 for wave equation, and -2.84 vs. -8.44 for the pendulum in Table 1. The Deep Galerkin method for the pendulum in complete physics *DGM $(\omega_0, \alpha)$*, being constrained by the equation, outperforms Neural ODE but is far inferior to APHYNITY models. In the incomplete physics case, *DGM $(\omega_0)$* fails to compensate for the missing information. The *incomplete physical models*, *Param PDE $(a, b)$* for the reaction-diffusion, *Param PDE $(c)$* for the wave equation, and *Param ODE $(\omega_0)$* and *Hamiltonian models* for the damped pendulum, have even poorer performances than purely data-driven ones, as can be expected since they ignore important dynamical components, e.g. friction in the pendulum case. Using APHYNITY with these imperfect physical models greatly improves forecasting accuracy in all cases, significantly outperforming purely data-driven models, and reaching results often close to the accuracy of the true ODE, when APHYNITY and the true ODE models are integrated with the same numerical scheme (which is different from the one used for data generation, hence the non-null errors even for the true equations), e.g. -5.92 vs. -5.24 for wave equation in

---

[3]This integration scheme is then different from the one used for data generation, the rationale for this choice being that when training a model one does not know how exactly the data has been generated.

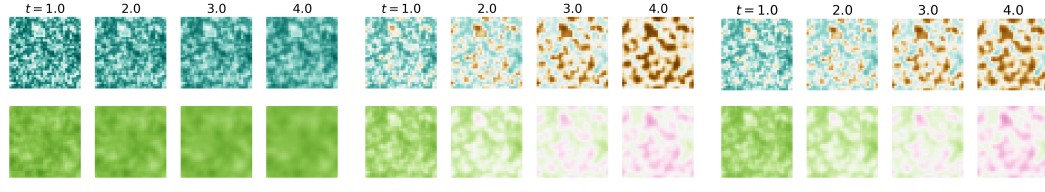

(a) Param PDE $(a, b)$, diffusion-only    (b) APHYNITY Param PDE $(a, b)$    (c) Ground truth simulation

Figure 2: Comparison of predictions of two components $u$ (top) and $v$ (bottom) of the reaction-diffusion system. Note that $t = 4$ is largely beyond the dataset horizon ($t = 2.5$).

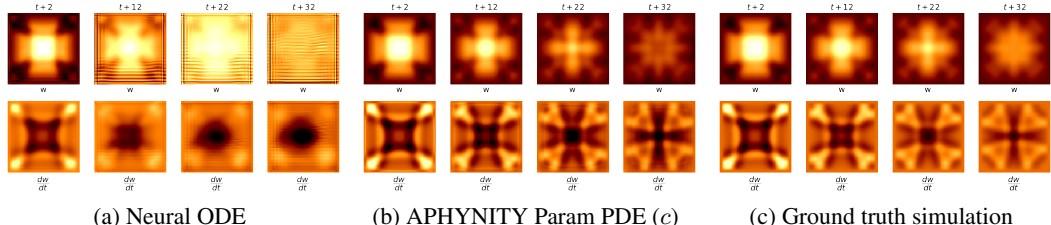

(a) Neural ODE      (b) APHYNITY Param PDE $(c)$      (c) Ground truth simulation

Figure 3: Comparison between the prediction of APHYNITY when $c$ is estimated and Neural ODE for the damped wave equation. Note that $t + 32$, last column for (a, b, c) is already beyond the training time horizon ($t + 25$), showing the consistency of APHYNITY method.

Table 1. This clearly highlights the capacity of our approach to augment incomplete physical models with a learned data-driven component.

**Physical parameter estimation** Confirming the phenomenon mentioned in the introduction and detailed in Appendix C, incomplete physical models can lead to bad estimates for the relevant physical parameters: an error respectively up to 67.6% and 10.4% for parameters in the reaction-diffusion and wave equations, and an error of more than 13% for parameters for the pendulum in Table 1. APHYNITY is able to significantly improve physical parameters identification: 2.3% error for the reaction-diffusion, 0.3% for the wave equation, and 4% for the pendulum. This validates the fact that augmenting a simple physical model to compensate its approximations is not only beneficial for prediction, but also helps to limit errors for parameter identification when dynamical models do not fit data well. This is crucial for interpretability and explainability of the estimates.

**Ablation study** We conduct ablation studies to validate the importance of the APHYNITY augmentation compared to a naive strategy consisting in learning $F = F_p + F_a$ without taking care on the quality of the decomposition, as done in (Wang et al., 2019; Mehta et al., 2020). Results shown in Table 1 of Appendix F show a consistent gain of APHYNITY for the three use cases and for all physical models: for instance for *Param ODE (a, b)* in reaction-diffusion, both forecasting performances ($\log \text{MSE}$ =-5.10 vs. -4.56) and identification parameter (Error= 2.33% vs. 6.39%) improve. Other ablation results are provided in Appendix F showing the relevance of the the trajectory-based approach described in Section 3.2 (vs supervising over finite difference approximations of the derivative $F$).

**Flexibility** When applied to complete physical models, APHYNITY does not degrade accuracy, contrary to a vanilla cooperation scheme (see ablations in Appendix F). This is due to the least action principle of our approach: when the physical knowledge is sufficient for properly predicting the observed dynamics, the model learns to ignore the data-driven augmentation. This is shown by the norm of the trained neural net component $F_a$, which is reported in Table 1 last column: as expected, $\|F_a\|^2$ diminishes as the complexity of the corresponding physical model increases, and, relative to incomplete models, the norm becomes very small for complete physical models (for example in the pendulum experiments, we have $\|F_a\| = 8.5$ for the APHYNITY model to be compared with 132 and 623 for the incomplete models). Thus, we see that the norm of $F_a$ is a good indication of how imperfect the physical models $\mathcal{F}_p$ are. It highlights the flexibility of APHYNITY to successfully adapt to very different levels of prior knowledge. Note also that APHYNITY sometimes slightly improves over the true ODE, as it compensates the error introduced by different numerical integration methods for data simulation and training (see Appendix E).

**Qualitative visualizations** Results in Figure 2 for reaction-diffusion show that the incomplete diffusion parametric PDE in Figure 2(a) is unable to properly match ground truth simulations: the

behavior of the two components in Figure 2(a) is reduced to simple independent diffusions due to the lack of interaction terms between $u$ and $v$. By using APHYNITY in Figure 2(b), the correlation between the two components appears together with the formation of Turing patterns, which is very similar to the ground truth. This confirms that $F_a$ can learn the reaction terms and improve prediction quality. In Figure 3, we see for the wave equation that the data-driven Neural ODE model fails at approximating $\mathrm{d}w/\mathrm{d}t$ as the forecast horizon increases: it misses crucial details for the second component $\mathrm{d}w/\mathrm{d}t$ which makes the forecast diverge from the ground truth. APHYNITY incorporates a Laplacian term as well as the data-driven $F_a$ thus capturing the damping phenomenon and succeeding in maintaining physically sound results for long term forecasts, unlike Neural ODE.

**Extension to non-stationary dynamics**   We provide additional results in Appendix G to tackle datasets where physical parameters of the equations vary in each sequence. To this end, we design an encoder able to perform parameter estimation for each sequence. Results show that APHYNITY accommodates well to this setting, with similar trends as those reported in this section.

**Additional illustrations**   We give further visual illustrations to demonstrate how the estimation of parameters in incomplete physical models is improved with APHYNITY. For the reaction-diffusion equation, we show that the incomplete parametric PDE underestimates both diffusion coefficients. The difference is visually recognizable between the poorly estimated diffusion (Figure 4(a)) and the true one (Figure 4(c)) while APHYNITY gives a fairly good estimation of those diffusion parameters as shown in Figure 4(b).

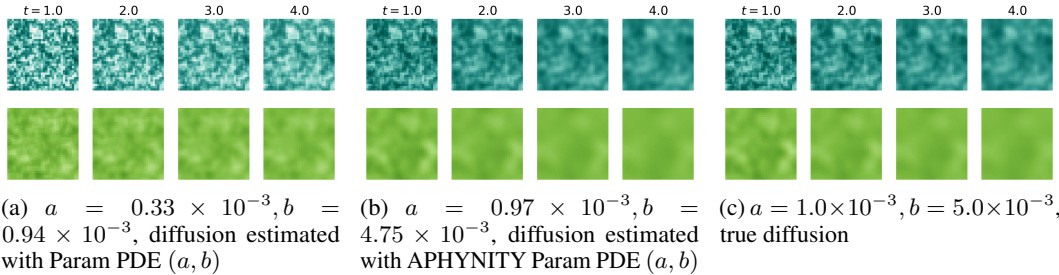

(a) $a = 0.33 \times 10^{-3}, b = 0.94 \times 10^{-3}$, diffusion estimated with Param PDE $(a, b)$

(b) $a = 0.97 \times 10^{-3}, b = 4.75 \times 10^{-3}$, diffusion estimated with APHYNITY Param PDE $(a, b)$

(c) $a = 1.0 \times 10^{-3}, b = 5.0 \times 10^{-3}$, true diffusion

Figure 4: Diffusion predictions using coefficient learned with (a) incomplete physical model Param PDE $(a, b)$ and (b) APHYNITY-augmented Param PDE$(a, b)$, compared with the (c) true diffusion

## 5   CONCLUSION

In this work, we introduce the APHYNITY framework that can efficiently augment approximate physical models with deep data-driven networks, performing similarly to models for which the underlying dynamics are entirely known. We exhibit the superiority of APHYNITY over data-driven, incomplete physics, and state-of-the-art approaches combining ML and MB methods, both in terms of forecasting and parameter identification on three various classes of physical systems. Besides, APHYNITY is flexible enough to adapt to different approximation levels of prior physical knowledge.

An appealing perspective is the applicability of APHYNITY on partially-observable settings, such as video prediction. Besides, we hope that the APHYNITY framework will open up the way to the design of a wide range of more flexible MB/ML models, e.g. in climate science, robotics or reinforcement learning. In particular, analyzing the theoretical decomposition properties in a partially-observed setting is an important direction for future work.

ACKNOWLEDGEMENTS:

Funding (P. Gallinari), Chaires de recherche et d'enseignement en intelligence artificielle (Chaires IA), DL4Clim project.

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

## A    REMINDER ON PROXIMINAL AND CHEBYSHEV SETS

We begin by giving a definition of proximinal and Chebyshev sets, taken from (Fletcher & Moors, 2014):

**Definition 1.** *A proximinal set of a normed space $(E, \|\cdot\|)$ is a subset $\mathcal{C} \subset E$ such that every $x \in E$ admits at least a nearest point in $\mathcal{C}$.*

**Definition 2.** *A Chebyshev set of a normed space $(E, \|\cdot\|)$ is a subset $\mathcal{C} \subset E$ such that every $x \in E$ admits a unique nearest point in $\mathcal{C}$.*

Proximinality reduces to a compacity condition in finite dimensional spaces. In general, it is a weaker one: Boundedly compact sets verify this property for example.

In Euclidean spaces, Chebyshev sets are simply the closed convex subsets. The question of knowing whether it is the case that all Chebyshev sets are closed convex sets in infinite dimensional Hilbert spaces is still an open question. In general, there exists examples of non-convex Chebyshev sets, a famous one being presented in (Johnson, 1987) for a non-complete inner-product space.

Given the importance of this topic in approximation theory, finding necessary conditions for a set to be Chebyshev and studying the properties of those sets have been the subject of many efforts. Some of those properties are summarized below:

- The metric projection on a boundedly compact Chebyshev set is continuous.
- If the norm is strict, every closed convex space, in particular any finite dimensional subspace is Chebyshev.
- In a Hilbert space, every closed convex set is Chebyshev.

## B    PROOF OF PROPOSITIONS 1 AND 2

We prove the following result which implies both propositions in the article:

**Proposition 3.** *The optimization problem:*

$$\min_{F_p \in \mathcal{F}_p, F_a \in \mathcal{F}} \quad \|F_a\| \quad \text{subject to} \quad \forall X \in \mathcal{D}, \forall t, \frac{\mathrm{d}X_t}{\mathrm{d}t} = (F_p + F_a)(X_t) \tag{5}$$

*is equivalent a metric projection onto $\mathcal{F}_p$.*

*If $\mathcal{F}_p$ is proximinal, Eq. (5) admits a minimizing pair.*

*If $\mathcal{F}_p$ is Chebyshev, Eq. (5) admits a unique minimizing pair which $F_p$ is the metric projection.*

*Proof.* The idea is to reconstruct the full functional from the trajectories of $\mathcal{D}$. By definition, $\mathcal{A}$ is the set of points reached by trajectories in $\mathcal{D}$ so that:

$$\mathcal{A} = \{x \in \mathbb{R}^d \mid \exists X. \in \mathcal{D}, \exists t, \ X_t = x\}$$

Then let us define a function $F^{\mathcal{D}}$ in the following way: For $a \in \mathcal{A}$, we can find $X. \in \mathcal{D}$ and $t_0$ such that $X_{t_0} = a$. Differentiating $X$ at $t_0$, which is possible by definition of $\mathcal{D}$, we take:

$$F^{\mathcal{D}}(a) = \left. \frac{\mathrm{d}X_t}{\mathrm{d}t} \right|_{t=t_0}$$

For any $(F_p, F_a)$ satisfying the constraint in Eq. (5), we then have that $(F_p + F_a)(a) = \mathrm{d}X_t/\mathrm{d}t_{|t_0} = F^{\mathcal{D}}(a)$ for all $a \in \mathcal{A}$. Conversely, any pair such that $(F_p, F_a) \in \mathcal{F}_p \times \mathcal{F}$ and $F_p + F_a = F^{\mathcal{D}}$, verifies the constraint.

Thus we have the equivalence between Eq. (5) and the metric projection formulated as:

$$\underset{F_p \in \mathcal{F}_p}{\text{minimize}} \quad \left\| F^{\mathcal{D}} - F_p \right\| \tag{6}$$

If $\mathcal{F}_p$ is proximinal, the projection problem admits a solution which we denote $F_p^\star$. Taking $F_a^\star = F^\mathcal{D} - F_p^\star$, we have that $F_p^\star + F_a^\star = F^\mathcal{D}$ so that $(F_p^\star, F_a^\star)$ verifies the constraint of Eq. (2). Moreover, if there is $(F_p, F_a)$ satisfying the constraint of Eq. (2), we have that $F_p + F_a = F^\mathcal{D}$ by what was shown above and $\|F_a\| = \|F^\mathcal{D} - F_p\| \geq \|F^\mathcal{D} - F_p^\star\|$ by definition of $F_p^\star$. This shows that $(F_p^\star, F_a^\star)$ is minimal.

Moreover, if $\mathcal{F}_p$ is a Chebyshev set, by uniqueness of the projection, if $F_p \neq F_p^\star$ then $\|F_a\| > \|F_a^\star\|$. Thus the minimal pair is unique.

$\square$

## C    PARAMETER ESTIMATION IN INCOMPLETE PHYSICAL MODELS

Classically, when a set $\mathcal{F}_p \subset \mathcal{F}$ summarising the most important properties of a system is available, this gives a *simplified model* of the true dynamics and the adopted problem is then to fit the trajectories using this model as well as possible, solving:

$$\begin{array}{ll} \underset{F_p \in \mathcal{F}_p}{\text{minimize}} & \mathbb{E}_{X \sim \mathcal{D}} L(\widetilde{X}^{X_0}, X) \\ \\ \text{subject to} & \forall g \in \mathcal{I}, \ \widetilde{X}_0^g = g \text{ and } \forall t, \ \frac{\mathrm{d}\widetilde{X}_t^g}{\mathrm{d}t} = F_p(\widetilde{X}_t^g) \end{array} \tag{7}$$

where $L$ is a discrepancy measure between trajectories. Recall that $\widetilde{X}^{X_0}$ is the result trajectory of an ODE solver taking $X_0$ as initial condition. In other words, we try to find a function $F_p$ which gives trajectories as close as possible to the ones from the dataset. While estimation of the function becomes easier, there is then a residual part which is left unexplained and this can be a non negligible issue in at least two ways:

- When $F \notin \mathcal{F}_p$, the loss is strictly positive at the minimum. This means that reducing the space of functions $\mathcal{F}_p$ makes us lose in terms of accuracy.[4]

- The obtained function $F_p$ might not even be the most meaningful function from $\mathcal{F}_p$ as it would try to capture phenomena which are not explainable with functions in $\mathcal{F}_p$, thus giving the wrong bias to the calculated function. For example, if one is considering a dampened periodic trajectory where only the period can be learned in $\mathcal{F}_p$ but not the dampening, the estimated period will account for the dampening and will thus be biased.

This is confirmed in the paper in Section 4: the incomplete physical models augmented with APHYNITY get different and experimentally better physical identification results than the physical models alone.

Let us compare our approach with this one on the linearized damped pendulum to show how estimates of physical parameters can differ. The equation is the following:

$$\frac{\mathrm{d}^2\theta}{\mathrm{d}t^2} + \omega_0^2 \theta + \alpha \frac{\mathrm{d}\theta}{\mathrm{d}t} = 0$$

We take the same notations as in the article and parametrize the simplified physical models as:

$$F_p^a : X \mapsto (\frac{\mathrm{d}\theta}{\mathrm{d}t}, -a\theta)$$

where $a > 0$ corresponds to $\omega_0^2$. The corresponding solution for an initial state $X_0$, which we denote $X^a$, can then written explicitly as:

$$\theta_t^a = \theta_0 \cos \sqrt{a} t$$

Let us consider damped pendulum solutions $X$ written as:

$$\theta_t = \theta_0 e^{-t} \cos t$$

which corresponds to:

$$F : X \mapsto (\frac{\mathrm{d}\theta}{\mathrm{d}t}, -2(\theta + \frac{\mathrm{d}\theta}{\mathrm{d}t}))$$

---

[4]This is true in theory, although not necessarily in practice when $F$ overfits a small dataset.

It is then easy to see that the estimate of $a$ with the physical model alone can be obtained by minimizing:

$$\int_0^T |e^{-t}\cos t - \cos\sqrt{a}t|^2$$

This expression depends on $T$ and thus, depending on the chosen time interval and the way the integral is discretized will almost always give biased estimates. In other words, the estimated value of $a$ will not give us the desired solution $t \mapsto \cos t$.

On the other hand, for a given $a$, in the APHYNITY framework, the residual must be equal to:

$$F_r^a : X \mapsto (0, (a-2)\theta - 2\frac{d\theta}{dt})$$

in order to satisfy the fitting constraint. Here $a$ corresponds to $1 + \omega_0^2$ not to $\omega_0^2$ as in the simplified case. Minimizing its norm, we obtain $a = 2$ which gives us the desired solution:

$$\theta_t = \theta_0 e^{-t}\cos t$$

with the right period.

## D    DISCUSSION ON SUPERVISION OVER DERIVATIVES

In order to find the appropriate decomposition $(F_p, F_a)$, we use a trajectory-based error by solving:

$$
\begin{aligned}
&\underset{F_p \in \mathcal{F}_p, F_a \in \mathcal{F}}{\text{minimize}} \quad \|F_a\| \\
&\text{subject to} \quad \forall g \in \mathcal{I}, \ \widetilde{X}_0^g = g \text{ and } \forall t, \ \frac{d\widetilde{X}_t^g}{dt} = (F_p + F_a)(\widetilde{X}_t^g), \\
&\qquad\qquad \forall X \in \mathcal{D}, \ L(X, \widetilde{X}^{X_0}) = 0
\end{aligned}
\tag{8}
$$

In the continuous setting where the data is available at all times $t$, this problem is in fact equivalent to the following one:

$$\underset{F_p \in \mathcal{F}_p}{\text{minimize}} \quad \mathbb{E}_{X \sim \mathcal{D}} \int \left\| \frac{dX_t}{dt} - F_p(X_t) \right\| \tag{9}$$

where the supervision is done directly over derivatives, obtained through finite-difference schemes. This echoes the proof in Section B of the Appendix where $F$ can be reconstructed from the continuous data.

However, in practice, data is only available at discrete times with a certain time resolution. While Eq. (9) is indeed equivalent to Eq. (8) in the continuous setting, in the practical discrete one, the way error propagates is not anymore: For Eq. (8) it is controlled over integrated trajectories while for Eq. (9) the supervision is over the approximate derivatives of the trajectories from the dataset. We argue that the trajectory-based approach is more flexible and more robust for the following reasons:

- In Eq. (8), if $F_a$ is appropriately parameterized, it is possible to perfectly fit the data trajectories at the sampled points.
- The use of finite differences schemes to estimate $F$ as is done in Eq. (9) necessarily induces a non-zero discretization error.
- This discretization error is explosive in terms of divergence from the true trajectories.

This last point is quite important, especially when time sampling is sparse (even though we do observe this adverse effect empirically in our experiments with relatively finely time-sampled trajectories). The following gives a heuristical reasoning as to why this is the case. Let $\widetilde{F} = F + \epsilon$ be the function estimated from the sampled points with an error $\epsilon$ such that $\|\epsilon\|_\infty \leq \alpha$. Denoting $\widetilde{X}$ the corresponding trajectory generated by $\widetilde{F}$, we then have, for all $X \in \mathcal{D}$:

$$\forall t, \ \frac{d(X - \widetilde{X})_t}{dt} = F(X_t) - F(\widetilde{X}_t) - \epsilon(\widetilde{X}_t)$$

Integrating over $[0, T]$ and using the triangular inequality as well as the mean value inequality, supposing that $F$ has uniformly bounded spatial derivatives:

$$\forall t \in [0, T], \ \|(X - \widetilde{X})_t\| \leq \|\nabla F\|_\infty \int_0^t \|X_s - \widetilde{X}_s\| + \alpha t$$

which, using a variant of the Grönwall lemma, gives us the inequality:

$$\forall t \in [0, T], \ \|X_t - \widetilde{X}_t\| \leq \frac{\alpha}{\|\nabla F\|_\infty}(\exp(\|\nabla F\|_\infty t) - 1)$$

When $\alpha$ tends to $0$, we recover the true trajectories $X$. However, as $\alpha$ is bounded away from $0$ by the available temporal resolution, this inequality gives a rough estimate of the way $\widetilde{X}$ diverges from them, and it can be an equality in many cases. This exponential behaviour explains our choice of a trajectory-based optimization.

## E  IMPLEMENTATION DETAILS

We describe here the three use cases studied in the paper for validating APHYNITY. All experiments are implemented with PyTorch (Paszke et al., 2019) and the differentiable ODE solvers with the adjoint method implemented in `torchdiffeq`.[5]

### E.1  REACTION-DIFFUSION EQUATIONS

The system is driven by a FitzHugh-Nagumo type PDE (Klaasen & Troy, 1984)

$$\frac{\partial u}{\partial t} = a\Delta u + R_u(u, v; k), \frac{\partial v}{\partial t} = b\Delta v + R_v(u, v)$$

where $a$ and $b$ are respectively the diffusion coefficients of $u$ and $v$, $\Delta$ is the Laplace operator. The local reaction terms are $R_u(u, v; k) = u - u^3 - k - v, R_v(u, v) = u - v$.

The state $X = (u, v)$ is defined over a compact rectangular domain $\Omega = [-1, 1]^2$ with periodic boundary conditions. $\Omega$ is spatially discretized with a $32 \times 32$ 2D uniform square mesh grid. The periodic boundary condition is implemented with circular padding around the borders. $\Delta$ is systematically estimated with a $3 \times 3$ discrete Laplace operator.

**Dataset**  Starting from a randomly sampled initial state $X_{\text{init}} \in [0, 1]^{2 \times 32 \times 32}$, we generate states by integrating the true PDE with fixed $a$, $b$, and $k$ in a dataset ($a = 1 \times 10^{-3}, b = 5 \times 10^{-3}, k = 5 \times 10^{-3}$). We firstly simulate high time-resolution ($\delta t_{\text{sim}} = 0.001$) sequences with explicit finite difference method. We then extract states every $\delta t_{\text{data}} = 0.1$ to construct our low time-resolution datasets.

We set the time of random initial state to $t = -0.5$ and the time horizon to $t = 2.5$. 1920 sequences are generated, with 1600 for training/validation and 320 for test. We take the state at $t = 0$ as $X_0$ and predict the sequence until the horizon (equivalent to 25 time steps) in all reaction-diffusion experiments. Note that the sub-sequence with $t < 0$ are reserved for the extensive experiments in Appendix G.1.

**Neural network architectures**  Our $F_a$ here is a 3-layer convolution network (ConvNet). The two input channels are $(u, v)$ and two output ones are $(\frac{\partial u}{\partial t}, \frac{\partial v}{\partial t})$. The purely data-driven Neural ODE uses such ConvNet as its $F$. The detailed architecture is provided in Table 2. The estimated physical parameters $\theta_p$ in $F_p$ are simply a trainable vector $(a, b) \in \mathbb{R}_+^2$ or $(a, b, k) \in \mathbb{R}_+^3$.

---

[5] https://github.com/rtqichen/torchdiffeq

Table 2: ConvNet architecture in reaction-diffusion and wave equation experiments, used as data-driven derivative operator in APHYNITY and Neural ODE (Chen et al., 2018).

| Module | Specification |
|---|---|
| 2D Conv. | $3 \times 3$ kernel, 2 input channels, 16 output channels, 1 pixel zero padding |
| 2D Batch Norm. | No average tracking |
| ReLU activation | — |
| 2D Conv. | $3 \times 3$ kernel, 16 input channels, 16 output channels, 1 pixel zero padding |
| 2D Batch Norm. | No average tracking |
| ReLU activation | — |
| 2D Conv. | $3 \times 3$ kernel, 16 input channels, 2 output channels, 1 pixel zero padding |

**Optimization hyperparameters**  We choose to apply the same hyperparameters for all the reaction-diffusion experiments: $Niter = 1, \lambda_0 = 1, \tau_1 = 1 \times 10^{-3}, \tau_2 = 1 \times 10^3$.

### E.2   WAVE EQUATIONS

The damped wave equation is defined by

$$\frac{\partial^2 w}{\partial t^2} - c^2 \Delta w + k \frac{\partial w}{\partial t} = 0$$

where $c$ is the wave speed and $k$ is the damping coefficient. The state is $X = (w, \frac{\partial w}{\partial t})$.

We consider a compact spatial domain $\Omega$ represented as a $64 \times 64$ grid and discretize the Laplacian operator similarly. $\Delta$ is implemented using a $5 \times 5$ discrete Laplace operator in simulation whereas in the experiment is a $3 \times 3$ Laplace operator. Null Neumann boundary condition are imposed for generation.

**Dataset**  $\delta t$ was set to $0.001$ to respect Courant number and provide stable integration. The simulation was integrated using a 4th order finite difference Runge-Kutta scheme for 300 steps from an initial Gaussian state, i.e for all sequence at $t = 0$, we have:

$$w(x, y, t = 0) = C \times \exp^{\frac{(x-x_0)^2 + (y-y_0)^2}{\sigma^2}} \tag{10}$$

The amplitude $C$ is fixed to 1, and $(x_0, y_0) = (32, 32)$ to make the Gaussian curve centered for all sequences. However, $\sigma$ is different for each sequence and uniformly sampled in $[10, 100]$. The same $\delta t$ was used for train and test. All initial conditions are Gaussian with varying amplitudes. 250 sequences are generated, 200 are used for training while 50 are reserved as a test set. In the main paper setting, $c = 330$ and $k = 50$. As with the reaction diffusion case, the algorithm takes as input a state $X_{t_0} = (w, \frac{dw}{dt})(t_0)$ and predicts all states from $t_0 + \delta t$ up to $t_0 + 25\delta t$.

**Neural network architectures**  The neural network for $F_a$ is a 3-layer convolution neural network with the same architecture as in Table 2. For $F_p$, the parameter(s) to be estimated is either a scalar $c \in \mathbb{R}_+$ or a vector $(c, k) \in \mathbb{R}_+^2$. Similarly, Neural ODE networks are build as presented in Table 2.

**Optimization hyperparameters**  We use the same hyperparameters for the experiments: $Niter = 3, \lambda_0 = 1, \tau_1 = 1 \times 10^{-4}, \tau_2 = 1 \times 10^2$.

### E.3   DAMPED PENDULUM

We consider the non-linear damped pendulum problem, governed by the ODE

$$\frac{d^2\theta}{dt^2} + \omega_0^2 \sin\theta + \alpha \frac{d\theta}{dt} = 0$$

where $\theta(t)$ is the angle, $\omega_0 = \frac{2\pi}{T_0}$ is the proper pulsation ($T_0$ being the period) and $\alpha$ is the damping coefficient. With the state $X = (\theta, \frac{d\theta}{dt})$, the ODE can be written as $\frac{dX_t}{dt} = F(X_t)$ with $F : X \mapsto (\frac{d\theta}{dt}, -\omega_0^2 \sin\theta - \alpha \frac{d\theta}{dt})$.

**Dataset** For each train / validation / test split, we simulate a dataset with 25 trajectories of 40 timesteps (time interval $[0, 20]$, timestep $\delta t = 0.5$) with fixed ODE coefficients ($T_0 = 12, \alpha = 0.2$) and varying initial conditions. The simulation integrator is Dormand-Prince Runge-Kutta method of order (4)5 (DOPRI5, Dormand & Prince, 1980). We also add a small amount of white gaussian noise ($\sigma = 0.01$) to the state. Note that our pendulum dataset is much more challenging than the ideal frictionless pendulum considered in Greydanus et al. (2019).

**Neural network architectures** We detail in Table 3 the neural architectures used for the damped pendulum experiments. All data-driven augmentations for approximating the mapping $X_t \mapsto F(X_t)$ are implemented by multi-layer perceptrons (MLP) with 3 layers of 200 neurons and ReLU activation functions (except at the last layer: linear activation). The Hamiltonian (Greydanus et al., 2019; Toth et al., 2020) is implemented by a MLP that takes the state $X_t$ and outputs a scalar estimation of the Hamiltonian $\mathcal{H}$ of the system: the derivative is then computed by an in-graph gradient of $\mathcal{H}$ with respect to the input: $F(X_t) = \left( \frac{\partial \mathcal{H}}{\partial (\mathrm{d}\theta / \mathrm{d}t)}, -\frac{\partial \mathcal{H}}{\mathrm{d}\theta} \right)$.

Table 3: Neural network architectures for the damped pendulum experiments. n/a corresponds to non-applicable cases.

| Method | Physical model | Data-driven model |
|---|---|---|
| Neural ODE | n/a | MLP(in=2, units=200, layers=3, out=2) |
| Hamiltonian | MLP(in=2, units=200, layers=3, out=1) | n/a |
| APHYNITY Hamiltonian | MLP(in=2, units=200, layers=3, out=1) | MLP(in=2, units=200, layers=3, out=2) |
| Param ODE ($\omega_0$) | 1 trainable parameter $\omega_0$ | n/a |
| APHYNITY Param ODE ($\omega_0$) | 1 trainable parameter $\omega_0$ | MLP(in=2, units=200, layers=3, out=2) |
| Param ODE ($\omega_0, \alpha$) | 2 trainable parameters $\omega_0, \lambda$ | n/a |
| APHYNITY Param ODE ($\omega_0, \alpha$) | 2 trainable parameters $\omega_0, \lambda$ | MLP(in=2, units=200, layers=3, out=2) |

**Optimization hyperparameters** The hyperparameters of the APHYNITY optimization algorithm ($Niter, \lambda_0, \tau_1, \tau_2$) were cross-validated on the validation set and are shown in Table 4. All models were trained with a maximum number of 5000 steps with early stopping.

Table 4: Hyperparameters of the damped pendulum experiments.

| Method | Niter | $\lambda_0$ | $\tau_1$ | $\tau_2$ |
|---|---|---|---|---|
| APHYNITY Hamiltonian | 5 | 1 | 1 | 0.1 |
| APHYNITY ParamODE ($\omega_0$) | 5 | 1 | 1 | 10 |
| APHYNITY ParamODE ($\omega_0, \lambda$) | 5 | 1000 | 1 | 100 |

# F  ABLATION STUDY

We conduct ablation studies to show the effectiveness of APHYNITY's adaptive optimization and trajectory-based learning scheme.

## F.1  ABLATION TO VANILLA MB/ML COOPERATION

In Table 5, we consider the ablation case with the vanilla augmentation scheme found in Le Guen & Thome (2020); Wang et al. (2019); Mehta et al. (2020), which does not present any proper decomposition guarantee. We observe that the APHYNITY cooperation scheme outperforms this vanilla scheme in all case, both in terms of forecasting performances (e.g. log MSE= -0.35 vs. -3.97 for the Hamiltonian in the pendulum case) and parameter identification (e.g. Err Param=8.4% vs. 2.3 for Param PDE ($a, b$ for reaction-diffusion). It confirms the crucial benefits of APHYNITY's principled decomposition scheme.

Table 5: Ablation study comparing APHYNITY to the vanilla augmentation scheme (Wang et al., 2019; Mehta et al., 2020) for the reaction-diffusion equation, wave equation and damped pendulum.

| Dataset | Method | log MSE | %Err Param. | $\|F_a\|^2$ |
|---|---|---|---|---|
| Reaction-diffusion | Param. PDE $(a, b)$ with vanilla aug. | -4.56±0.52 | 8.4 | (7.5±1.4)e1 |
| | APHYNITY Param. PDE $(a, b)$ | **-5.10±0.21** | **2.3** | (6.7±0.4)e1 |
| | Param. PDE $(a, b, k)$ with vanilla aug. | -8.04±0.03 | 25.4 | (1.5±0.2)e-2 |
| | APHYNITY Param. PDE $(a, b, k)$ | **-9.35±0.02** | **0.096** | (1.5±0.4)e-6 |
| | True PDE with vanilla aug. | -8.12±0.05 | n/a | (6.1±2.3)e-4 |
| | APHYNITY True PDE | **-9.17±0.02** | n/a | (1.4±0.8)e-7 |
| Wave equation | Param PDE $(c)$ with vanilla aug. | -3.90 ± 0.27 | 0.51 | 88.66 |
| | APHYNITY Param PDE $(c)$ | **-4.64±0.25** | **0.31** | 71.0 |
| | Param PDE $(c, k)$ with vanilla aug. | -5.96 ± 0.10 | 0.71 | 25.1 |
| | APHYNITY Param PDE $(c, k)$ | **-6.09±0.28** | **0.70** | 4.54 |
| Damped pendulum | Hamiltonian with vanilla aug. | -0.35±0.1 | n/a | 837±117 |
| | APHYNITY Hamiltonian | **-3.97±1.2** | n/a | 623±68 |
| | Param ODE $(\omega_0)$ with vanilla aug. | -7.02±1.7 | 4.5 | 148±49 |
| | APHYNITY Param ODE $(\omega_0)$ | **-7.86±0.6** | **4.0** | 132±11 |
| | Param ODE $(\omega_0, \alpha)$ with vanilla aug. | -7.60±0.6 | 4.65 | 35.5±6.2 |
| | APHYNITY Param ODE $(\omega_0, \alpha)$ | **-8.31±0.3** | **0.39** | 8.5±2.0 |
| | Augmented True ODE with vanilla aug. | **-8.40±0.2** | n/a | 3.4±0.8 |
| | APHYNITY True ODE | **-8.44±0.2** | n/a | 2.3±0.4 |

## F.2 DETAILED ABLATION STUDY

We conduct also two other ablations in Table 6:

- *derivative supervision*: in which $F_p + F_a$ is trained with supervision over approximated derivatives on ground truth trajectory, as performed in Greydanus et al. (2019); Cranmer et al. (2020). More precisely, APHYNITY's $\mathcal{L}_{\text{traj}}$ is here replaced with $\mathcal{L}_{\text{deriv}} = \|\frac{\mathrm{d}X_t}{\mathrm{d}t} - F(X_t)\|$ as in Eq. (9), where $\frac{\mathrm{d}X_t}{\mathrm{d}t}$ is approximated by finite differences on $X_t$.

- *non-adaptive optim.*: in which we train APHYNITY by minimizing $\|F_a\|$ without the adaptive optimization of $\lambda$ shown in Algorithm 1. This case is equivalent to $\lambda = 1, \tau_2 = 0$.

We highlight the importance to use a principled adaptive optimization algorithm (APHYNITY algorithm described in paper) compared to a non-adpative optimization: for example in the reaction-diffusion case, log MSE= -4.55 vs. -5.10 for Param PDE $(a, b)$. Finally, when the supervision occurs on the derivative, both forecasting and parameter identification results are systematically lower than with APHYNITY's trajectory based approach: for example, log MSE=-1.16 vs. -4.64 for Param PDE $(c)$ in the wave equation. It confirms the good properties of the APHYNITY training scheme.

Table 6: Detailed ablation study on supervision and optimization for the reaction-diffusion equation, wave equation and damped pendulum.

| Dataset | Method | log MSE | %Err Param. | $\|F_a\|^2$ |
|---|---|---|---|---|
| Reaction-diffusion | Augmented Param. PDE $(a, b)$ derivative supervision | -4.42±0.25 | 12.6 | (6.8±0.6)e1 |
| | Augmented Param. PDE $(a, b)$ non-adaptive optim. | -4.55±0.11 | 7.5 | (7.6±1.0)e1 |
| | APHYNITY Param. PDE $(a, b)$ | **-5.10±0.21** | **2.3** | (6.7±0.4)e1 |
| | Augmented Param. PDE $(a, b, k)$ derivative supervision | -4.90±0.06 | 11.7 | (1.9±0.3)e-1 |
| | Augmented Param. PDE $(a, b, k)$ non-adaptive optim. | -9.10±0.02 | 0.21 | (5.5±2.9)e-7 |
| | APHYNITY Param. PDE $(a, b, k)$ | **-9.35±0.02** | **0.096** | (1.5±0.4)e-6 |
| | Augmented True PDE derivative supervision | -6.03±0.01 | n/a | (3.1±0.8)e-3 |
| | Augmented True PDE non-adaptive optim. | -9.01±0.01 | n/a | (1.5±0.8)e-6 |
| | APHYNITY True PDE | **-9.17±0.02** | n/a | (1.4±0.8)e-7 |
| Wave equation | Augmented Param PDE $(c)$ derivative supervision | -1.16±0.48 | 12.1 | 0.00024 |
| | Augmented Param PDE $(c)$ non-adaptive optim. | -2.57±0.21 | 3.1 | 43.6 |
| | APHYNITY Param PDE $(c)$ | **-4.64±0.25** | **0.31** | 71.0 |
| | Augmented Param PDE $(c, k)$ derivative supervision | -4.19±0.36 | 7.2 | 0.00012 |
| | Augmented Param PDE $(c, k)$ non-adaptive optim. | -4.93±0.51 | 1.32 | 0.054 |
| | APHYNITY Param PDE $(c, k)$ | **-6.09±0.28** | **0.70** | 4.54 |
| | Augmented True PDE derivative supervision | -4.42 ± 0.33 | n/a | 6.02e-5 |
| | Augmented True PDE non-adaptive optim. | -4.97±0.49 | n/a | 0.23 |
| | APHYNITY True PDE | **-5.24±0.45** | n/a | 0.14 |
| Damped pendulum | Augmented Hamiltonian derivative supervision | -0.83±0.3 | n/a | 642±121 |
| | Augmented Hamiltonian non-adaptive optim. | -0.49±0.58 | n/a | 165±30 |
| | APHYNITY Hamiltonian | **-3.97±1.2** | n/a | 623±68 |
| | Augmented Param ODE $(\omega_0)$ derivative supervision | -1.02±0.04 | 5.8 | 136±13 |
| | Augmented Param ODE $(\omega_0)$ non-adaptive optim. | -4.30±1.3 | 4.4 | 90.4±27 |
| | APHYNITY Param ODE $(\omega_0)$ | **-7.86±0.6** | **4.0** | 132±11 |
| | Augmented Param ODE $(\omega_0, \alpha)$ derivative supervision | -2.61±0.2 | 5.0 | 3.2±1.7 |
| | Augmented Param ODE $(\omega_0, \alpha)$ non-adaptive optim. | -7.69±1.3 | 1.65 | 4.8±7.7 |
| | APHYNITY Param ODE $(\omega_0, \alpha)$ | **-8.31±0.3** | **0.39** | 8.5±2.0 |
| | Augmented True ODE derivative supervision | -2.14±0.3 | n/a | 4.1±0.6 |
| | Augmented True ODE non-adaptive optim. | **-8.34±0.4** | n/a | 1.4±0.3 |
| | APHYNITY True ODE | **-8.44±0.2** | n/a | 2.3±0.4 |

# G ADDITIONAL EXPERIMENTS

## G.1 REACTION-DIFFUSION SYSTEMS WITH VARYING DIFFUSION PARAMETERS

We conduct an extensive evaluation on a setting with varying diffusion parameters for reaction-diffusion equations. The only varying parameters are diffusion coefficients, i.e. individual $a$ and $b$ for each sequence. We randomly sample $a \in [1 \times 10^{-3}, 2 \times 10^{-3}]$ and $b \in [3 \times 10^{-3}, 7 \times 10^{-3}]$. $k$ is still fixed to $5 \times 10^{-3}$ across the dataset.

In order to estimate $a$ and $b$ for each sequence, we use here a ConvNet encoder $E$ to estimate parameters from 5 reserved frames ($t < 0$). The architecture of the encoder $E$ is similar to the one in Table 2 except that $E$ takes 5 frames (10 channels) as input and $E$ outputs a vector of estimated $(\tilde{a}, \tilde{b})$ after applying a sigmoid activation scaled by $1 \times 10^{-2}$ (to avoid possible divergence). For the baseline Neural ODE, we concatenate $a$ and $b$ to each sequence as two channels.

In Table 7, we observe that combining data-driven and physical components outperforms the pure data-driven one. When applying APHYNITY to Param PDE $(a, b)$, the prediction precision is significantly improved (log MSE: -1.32 vs. -4.32) with $a$ and $b$ respectively reduced from 55.6% and 54.1% to 11.8% and 18.7%. For complete physics cases, the parameter estimations are also improved for Param PDE $(a, b, k)$ by reducing over 60% of the error of $b$ (3.10 vs. 1.23) and 10% to 20% of the errors of $a$ and $k$ (resp. 1.55/0.59 vs. 1.29/0.39).

The extensive results reflect the same conclusion as shown in the main article: APHYNITY improves the prediction precision and parameter estimation. The same decreasing tendency of $\|F_a\|$ is also confirmed.

Table 7: Results of the dataset of reaction-diffusion with varying $(a, b)$. $k = 5 \times 10^{-3}$ is shared across the dataset.

| | Method | log MSE | %Err $a$ | %Err $b$ | %Err $k$ | $\|F_a\|^2$ |
|---|---|---|---|---|---|---|
| Data-driven | Neural ODE (Chen et al., 2018) | -3.61±0.07 | n/a | n/a | n/a | n/a |
| Incomplete physics | Param PDE $(a, b)$ | -1.32±0.02 | 55.6 | 54.1 | n/a | n/a |
| | APHYNITY Param PDE $(a, b)$ | **-4.32±0.32** | **11.8** | **18.7** | n/a | (4.3±0.6)e1 |
| Complete physics | Param PDE $(a, b, k)$ | **-5.54±0.38** | 1.55 | 3.10 | 0.59 | n/a |
| | APHYNITY Param PDE $(a, b, k)$ | **-5.72±0.25** | **1.29** | **1.23** | **0.39** | (5.9±4.3)e-1 |
| | True PDE | **-8.86±0.02** | n/a | n/a | n/a | n/a |
| | APHYNITY True PDE | **-8.82±0.15** | n/a | n/a | n/a | (1.8±0.6)e-5 |

## G.2 ADDITIONAL RESULTS FOR THE WAVE EQUATION

We conduct an experiment where each sequence is generated with a different wave celerity. This dataset is challenging because both $c$ and the initial conditions vary across the sequences. For each simulated sequence, an initial condition is sampled as described previously, along with a wave celerity $c$ also sampled uniformly in $[300, 400]$. Finally our initial state is integrated with the same Runge-Kutta scheme. 200 of such sequences are generated for training while 50 are kept for testing.

For this experiment, we also use a ConvNet encoder to estimate the wave speed $c$ from 5 consecutive reserved states $(w, \frac{\partial w}{\partial t})$. The architecture of the encoder $E$ is the same as in Table 2 but with 10 input channels. Here also, $k$ is fixed for all sequences and $k = 50$. The hyper-parameters used in these experiments are the same than described in the Section E.2.

The results when multiple wave speeds $c$ are in the dataset are consistent with the one present when only one is considered. Indeed, while prediction performances are slightly hindered, the parameter estimation remains consistent for both $c$ and $k$. This extension provides elements attesting for the robustness and adaptability of our method to more complex settings. Finally the purely data-driven Neural-ODE fails to cope with the increasing difficulty.

Table 8: Results for the damped wave equation when considering multiple $c$ sampled uniformly in $[300, 400]$ in the dataset, $k$ is shared across all sequences and $k = 50$.

| | Method | log MSE | %Error $c$ | %Error $k$ | $\|F_a\|^2$ |
|---|---|---|---|---|---|
| Data-driven | Neural ODE | 0.056±0.34 | n/a | n/a | n/a |
| Incomplete physics | Param PDE $(c)$ | -1.32±0.27 | 23.9 | n/a | n/a |
| | APHYNITY Param PDE $(c)$ | **-4.51±0.38** | 3.2 | n/a | 171 |
| Complete physics | Param PDE $(c, k)$ | -4.25±0.28 | 3.54 | 1.43 | n/a |
| | APHYNITY Param PDE $(c, k)$ | **-4.84±0.57** | 2.41 | 0.064 | 3.64 |
| | True PDE $(c, k)$ | **-4.51±0.29** | n/a | n/a | n/a |
| | APHYNITY True PDE $(c, k)$ | **-4.49±0.22** | n/a | n/a | 0.0005 |

## G.3 DAMPED PENDULUM WITH VARYING PARAMETERS

To extend the experiments conducted in the paper (section 4) with fixed parameters ($T_0 = 6, \alpha = 0.2$) and varying initial conditions, we evaluate APHYNITY on a much more challenging dataset where we vary both the parameters ($T_0, \alpha$) and the initial conditions between trajectories.

We simulate 500/50/50 trajectories for the train/valid/test sets integrated with DOPRI5. For each trajectory, the period $T_0$ (resp. the damping coefficient $\alpha$) are sampled uniformly in the range $[3, 10]$ (resp. $[0, 0.5]$).

We train models that take the first 20 steps as input and predict the next 20 steps. To account for the varying ODE parameters between sequences, we use an encoder that estimates the parameters based

on the first 20 timesteps. In practice, we use a recurrent encoder composed of 1 layer of 128 GRU units. The output of the encoder is fed as additional input to the data-driven augmentation models and to an MLP with final softplus activations to estimate the physical parameters when necessary ($\omega_0 \in \mathbb{R}_+$ for Param ODE ($\omega_0$), ($\omega_0, \alpha) \in \mathbb{R}_+^2$ for Param ODE ($\omega_0, \alpha$)).

In this varying ODE context, we also compare to the state-of-the-art univariate time series forecasting method N-Beats (Oreshkin et al., 2020).

Results shown in Table 9 are consistent with those presented in the paper. Pure data-driven models Neural ODE (Chen et al., 2018) and N-Beats (Oreshkin et al., 2020) fail to properly extrapolate the pendulum dynamics. Incomplete physical models (Hamiltonian and ParamODE ($\omega_0$)) are even worse since they do not account for friction. Augmenting them with APHYNITY significantly and consistently improves forecasting results and parameter identification.

Table 9: Forecasting and identification results on the damped pendulum dataset with different parameters for each sequence. log MSEs are computed over 20 predicted time-steps. For each level of incorporated physical knowledge, equivalent best results according to a Student t-test are shown in bold. n/a corresponds to non-applicable cases.

| | Method | log MSE | %Error $T_0$ | %Error $\alpha$ | $\|F_a\|^2$ |
|---|---|---|---|---|---|
| data-driven | Neural ODE (Chen et al., 2018) | -4.35±0.9 | n/a | n/a | n/a |
| | N-Beats (Oreshkin et al., 2020) | -4.57±0.5 | n/a | n/a | n/a |
| Incomplete physics | Hamiltonian (Greydanus et al., 2019) | -1.31±0.4 | n/a | n/a | n/a |
| | APHYNITY Hamiltonian | **-4.72±0.4** | n/a | n/a | 5.6±0.6 |
| | Param ODE ($\omega_0$) | -2.66±0.9 | 21.5±19 | n/a | n/a |
| | APHYNITY Param ODE ($\omega_0$) | **-5.94±0.7** | **5.0±1.8** | n/a | 0.49±0.1 |
| Complete physics | Param ODE ($\omega_0, \alpha$) | **-5.71±0.4** | 4.08±0.8 | 152±129 | n/a |
| | APHYNITY Param ODE ($\omega_0, \alpha$) | **-6.22±0.7** | **3.26±0.6** | **62±27** | (5.39±0.1)e-10 |
| | True ODE | **-8.58±0.1** | n/a | n/a | n/a |
| | APHYNITY True ODE | **-8.58±0.1** | n/a | n/a | (2.15±1.6)e-4 |

