# OpenReview forum: "Augmenting Physical Models with Deep Networks for Complex Dynamics Forecasting"
_ICLR.cc/2021/Conference — ICLR 2021 Oral_

### Official Review · AnonReviewer1 · 2020-10-22
**Sound work**

**Rating:** 8
**Confidence:** 4

**Review:**

### Summary of my understanding

The authors propose a method of function fitting for differential equations. They premise that a model $F$ is the additive combination of $F_p$ and $F_a$, which denote physics and augmenting parts, respectively. The functional form of the physics part, $F_p$, is given in accordance with prior knowledge, whereas the augmenting part, $F_a$, is modeled by neural nets. The proposed method follows the principle of least action of $F_a$, and the authors suggest solving a constrained optimization problem via a method of Lagrange multipliers. They show numerical results on three PDE/ODE-governed systems.

### Evaluation

I really enjoyed reading the paper, which is well written. The motivation is clearly presented. The related work can be more detailed given the recent active studies on physics + ML but seems sufficient from the viewpoint of ODE/PDE fitting. The proposed method is simple yet reasonable, and the experiments are enough supportive to see the superiority of the method. The ablation study in Appendix F is also very interesting. Possible improvement, which the authors would be aware of, can be found in the lack of experiments on real-world datasets. (I understand the difficulty of finding illustrative real-world examples in this kind of problem, but I couldn't stop pointing out it.) Overall, I think this is certainly sound work.

---

> ### Author Response · Authors · 2020-11-17
> **Response to Reviewer 1**
>
> Thank you for your comments and positive feedback! We will add a more detailed discussion of relevant related works with the additional ninth page available for the revised version of the paper.
>
> Moreover, regarding experimentation on real-world datasets, we agree that it is as important a step as it is difficult to achieve. When dealing with such data, many issues arise which are orthogonal to the properties of the model we are presenting here, including structured noise in observations, partial observability of the system, varying boundary conditions, non-stationary dynamics,... The experiments we present here already show that APHYNITY is robust and can handle different dynamics, some of which are quite challenging, successfully. The (preliminary) experiments with a toy partially observable video prediction task tend to confirm that the model can be extended to more realistic settings and this will be the subject of our future efforts. We would also be very happy to see different communities use APHYNITY for their application domains!

---

### Official Review · AnonReviewer2 · 2020-10-31
**tentative accept**

**Rating:** 7
**Confidence:** 3

**Review:**

Summary:
This paper outlines a method for forecasting and parameter estimation when you have a partial physics model (possibly with unknown parameters) and time series data. This is a hybrid approach where the data-driven (deep learning) approach only learns the parts not accounted for by the physical model. A key feature is being able to decompose the problem in such a way that the data-driven model only models what cannot be captured by the physical model. The parameters of these two models must be fit jointly so that the physical model's parameters are more correct. They prove existence and uniqueness for this decomposition.

Strong points:
I think this point of making the decomposition unique (and thus better fitting the physical model's parameters) is intuitive and a good one. To my knowledge, other work on hybrid models do not have theory showing this. (But I haven't extensively read the literature.)

The authors included extensive experiments with multiple versions of each of three test cases, plus several baselines. They also conducted ablation studies for three features of their method:
- using a decomposition to ensure that the data-driven model only learns what the physical model cannot contribute
- the trajectory (integral) approach instead of training on estimated derivatives
- using adaptive weights in the objective function

Weak points:
I think that the claim "We show that APHYNITY can perform similarly to complete physical models by augmenting incomplete ones, both in forecasting accuracy and physical parameter identification." is overstated. I compared APHYNITY for incomplete physics to the Param PDE or True PDE with complete physics in Table 1, and this claim seems true for the wave equation only.

I think that the related work section should acknowledge that people have been using neural networks for hybrids with physical models since the 1990s. Example: [1]. This is sometimes called "gray box" modeling or "hybrid" modeling. Since this section only mentions very recent work, I'm not very confident in the paper's claim that they are the first to have a principled decomposition with existence & uniqueness.

I think that "Propositions 1 and 2 provide, under mild conditions, the theoretical guarantees for the APHYNITY formulation to infer the correct MB/ML decomposition, thus enabling both recovering the proper physical parameters and accurate forecasting." is overstated. The theory is about the decomposition existing & being unique, not about this method being able to find that decomposition.
1. The theory makes no assumptions about the form of F_a (the data-driven component). Showing that the decomposition is possible is not the same as showing that there exists a neural network that could represent F_a. The different versions of the universal approximation theorem for neural networks that I've seen all require that the neural network is approximating something reasonably well-behaved (such as continuous). I could imagine that the parts of a dataset that cannot be modeled by an ODE/PDE might also be non-continuous, etc.
2. Further, even if the universal approximation theorem applied here, that would show that such a neural network exists, not how well any existing training methods can find that neural network.

I think "as expected, ||F_a||^2 diminishes as the complexity of the corresponding physical model increases, and becomes almost null for complete physical models." is too strong. It does decrease, but I wouldn't call 0.14 and 2.3 "almost null."

I personally can't vouch for the theory (in Section 3.1 & Appendix A & B), as I'm having a hard time following it. I hope that another reviewer can. Some things that would help me:
- The intuition would be easier if it's mentioned earlier what \mathcal{F} is in this paper's examples. L^2? It would be nice to put it before Section 3.1, right when the notation is introduced. Similarly, it would help if an example of \mathcal{F}_p was given early on.
- Some examples are given in Section 3.1 & the appendix of conditions that make a set Chebyshev. However, since I was having a hard time knowing which sets & spaces were important in this paper, I was having a hard time understanding which conditions were important.

Since reviewers aren't required to read the appendix, and it's quite long, I read it less closely. However, it contains the proofs and backs up many of the nice claims in the paper. This means that quite a bit of content in the paper is less vetted.



Other clarification questions:

I may have missed it, but does the paper describe how parameters are fit for the PDEs (physical models)? Example: ParamPDE(a,b) for reaction-diffusion.

Does the requirement of \mathcal{F}_p being Chebyshev mean that the parameters of the ODE/PDE need to be closed on one end? In other words, we couldn't search for a parameter in (-infinity, infinity)?

Are the results in Table 1 test errors? (as opposed to training or validation?)

It's not clear to me what you mean by the "partially observable" setting.

For the ablation studies, we're presumably comparing these variants to the full method's results, as described in Table 1, right? However, there are some discrepancies between Table 1 and Tables 5 & 6. Some of the numbers don't match and one of the wave equation cases is not included in Table 5. The "Augmented True PDE derivative supervision" case is also missing for the wave equation in Table 6.


Minor points:

It seems that the definition of the set of observed trajectories (in the beginning of Section 3) is not right.
- Your observations are for a finite set of values t in [0,T].
- Your observations are not across all solutions X for the dynamics: just at a finite number of IC/BCs.
- For the reaction-diffusion & wave equations, X is also a function of spatial x, and the data is collected at finitely many spatial points.

I think instead of defining a set \mathcal{A} on page 3, it would be clearer to use the space \mathbb{R}^d? I don't think F maps into just the set A, where A is defined as the set of values that X takes. Couldn't the derivatives have values that F doesn't reach?

On the bottom of Page 3, the question of the minimum being well-defined is explained as the existence question. However, I'm used to "well-defined" in various mathematical contexts meaning roughly "a unique answer." Is there a different meaning of "well-defined" in this context?

Summary: I tentatively recommend accepting this paper. Their method has quite a few empirical results showing improvements. People are interested in combining physical models with machine learning in a variety of scientific application areas, so I could see this being a well-used method. However, I have listed some points above that I would like fixed or clarified. I also hope that someone else can vouch for the proof.


[1] Rico-Martinez, R., J. S. Anderson, and I. G. Kevrekidis. "Continuous-time nonlinear signal processing: a neural network based approach for gray box identification." In Proceedings of IEEE Workshop on Neural Networks for Signal Processing, pp. 596-605. IEEE, 1994.

---

> ### Author Response · Authors · 2020-11-17
> **Response to Reviewer 2 (3/3): other clarifications**
>
> Below we answer your other clarification questions:
>
> **"I may have missed it, but does the paper describe how parameters are fit for the PDEs (physical models)? Example: ParamPDE(a,b) for reaction-diffusion."**
> The unknown parameters of physical models are defined as learnable parameters, such that it can be integrated into the computation graph. Then these parameters are fit with gradient descent by back-propagating the prediction error.
>
> **"Does the requirement of Fp being Chebyshev mean that the parameters of the ODE/PDE need to be closed on one end? In other words, we couldn’t search for a parameter in (-infinity,infinity)? "**
>  Closedness (in the topological sense) is indeed a necessary condition, otherwise the optimal $F_p$ could lie outside $\mathcal{F}_p$ (albeit certainly in its closure). In particular, if we are looking for a set defined by a parameter from a bounded interval of $\mathbb{R}$, this interval would have to include its boundaries. However, intervals of the form $(-\infty,+\infty)$, $(-\infty,A]$ or $[A,+\infty)$ are all closed sets. Moreover, $\mathcal{F}_p$ doesn't have to be bounded.
>
> **"Are the results in Table 1 test errors? (as opposed to training or validation?)"**
> The results reported are indeed test errors.
>
> **"It’s not clear to me what you mean by the "partially observable" setting."**
> The partially-observed setting is when the full state $X_t$ describing the system is not available, but we have instead "observations" $Y_t = g(X_t)$, where $g$ describes the loss of information and can be for example a masking function (e.g. observing only $\theta(t)$ and not the full state $(\theta(t), \frac{d\theta(t)}{d t})$ for the pendulum). This is a very important setting for many practical application areas such as climate sciences, robotics,...
>
> **"For the ablation studies, we’re presumably comparing these variants to the full method’s results, as described in Table 1, right? However, there are some discrepancies between Table1 and Tables 5 & 6. Some of the numbers don’t match and one of the wave equation cases is not included in Table 5. The "Augmented True PDE derivative supervision" case is also missing for the wave equation in Table 6."**
> You are right, the comparisons are made w.r.t. to the full APHYNITY model presented in Table 1. Thank you for pointing out the small discrepancies in reported results, we have checked each table and unified the results in the revised version. The missing line will also be added. We apologize for the discrepancies which are due to different round-off conventions and copying errors.
>
> **"It seems that the definition of the set of observed trajectories..at finitely many spatial points."**
> The set $\mathcal{D}$ is the set of all trajectories which are theoretically available and contains an infinite number of trajectories and as well as an infinite number of points for each trajectory. Obviously, as you point out, the dataset which is actually used for training, $\mathcal{D}_{\text{train}}$ is a discretized version of it, with a finite number of samples (both in terms of trajectories and of spatial and temporal resolution for each trajectory).
>
> **"I think instead of defining a set $\mathcal{A}$ on page 3, it would be clearer to use the space $\mathbb{R}^d$? I don't think F maps into just the set A, where A is defined as the set of values that X takes. Couldn't the derivatives have values that F doesn't reach?"**
> You are right, this is an error we have overlooked: $F$ does have $\mathcal{A}$ as its domain but its range is not constrained and should indeed be $\mathbb{R}^d$ (or the tangent bundle of $\mathcal{A}$ if we generalize our formalism to manifold-constrained dynamics).
>
> **"On the bottom of Page 3, the question of the minimum being well-defined is explained as the existence question. However, I’m used to "well-defined" in various mathematicalcontexts meaning roughly "a unique answer." Is there a different meaning of "well-defined"in this context?"**
> Here, "well-defined" refers to the optimal value of the optimization problem which is shown in Proposition 1.

---

> ### Author Response · Authors · 2020-11-17
> **Response to Reviewer 2 (2/3) : modeling and training**
>
> Below we answer your questions on the modeling assumptions and training of APHYNITY:
>
> **"The theory makes no assumptions about the form of $F_a$ (the data-driven component). [...] I could imagine that the parts of a dataset that cannot be modeled by an ODE/PDE might also be non-continuous, etc."**
>  The set of continuous functions over a domain are dense in many spaces of interest, which means for example that neural networks can indeed approximate $L^p$ spaces under reasonable assumptions and w.r.t. the corresponding norms. For example, Theorem 1 of [7] shows that bounded width neural networks can approximate any integrable function in $L^1$. In particular, piece-wise regular functions can be approximated with neural networks and it is thus a reasonable parametrization to choose.
>
>   However, there can obviously be no uniform convergence for neural networks towards discontinuous functions as they give continuous approximations of those functions. An interesting idea to explore would be to see whether using discontinuous neural networks (e.g. with discontinuous activation functions) would help in estimating dynamics which present many irregularities. In practice, for the dynamics we have studied, the expressive power of neural networks hasn't been an obstacle in our experiments.
>
>   Moreover, let us note that while we do propose a neural implementation of APHYNITY which works quite well in our experiments, other parametric families for $F_a$ might be relevant in other cases, the only assumption being that they should be able to recover a good approximation of any function in $\mathcal{F}$.
>
> **"Further, even if the universal approximation theorem applied here, that would show thatsuch a neural network exists, not how well any existing training methods can find that neural network."**
>  Indeed, when modelling $F_a$ with deep neural networks, we have no convergence guarantees to the global optimum due to the non-convexity of the objective, which is a common problem in deep learning. However, it has been noticed that local minima are not a problem in practice [9,4] and neural networks behave very well, often achieving zero training loss [8] and generalizing  well on unseen data points [5, 6]. The properties of deep overparametrized neural nets when optimized with modern SGD methods are still not fully understood and we do not have guarantees of convergence to the optimal decomposition: this is the cost of working with a deep neural parametrization. However, we find it valuable nonetheless to prove theoretical properties of the optimization problem being solved, independently from the practical parametrization and implementation which approximate it. Moreover, our experiments are consistent with this theory, for example regarding the accuracy of identified physical parameters, the decreasing norms of $F_a$ with increasing prior knowledge, the enhanced forecasting accuracy,...
>
> **"I think "as expected, $||F_a||^2$ diminishes as the complexity of the corresponding physical model increases, and becomes almost null for complete physical models." is too strong. It does decrease, but I wouldn't call 0.14 and 2.3 "almost null.""**
>   Actually, the absolute values of the residual norm cannot be compared across the experiments as the domains over which we sum are different. One should rather compare the relative differences between the incomplete and complete models: for example, $2.3$ is to be compared to the $632$ and $123$ norms obtained for incomplete models, thus amounting to, respectively, $0.4\%$ and $1.9\%$ which we judge to be quite close to zero and which account to discretization errors. The most important aspect is, as you note, the fact that the values do decrease as expected with the increasing completeness of the model. We will add this discussion to the revised version of the paper.
>
>  [4] Global optimality in tensor factorization, deep learning, and beyond - Haeffele, B. D. and Vidal, R. (2015).
>
>    [5]: Zhang, Chiyuan, et al; Understanding Deep Learning Requires Rethinking Generalization; 2016; https://openreview.net/forum?id=Sy8gdB9xx.
>
>    [6]: Belkin, Mikhail, et al. « Reconciling Modern Machine-Learning Practice and the Classical Bias–Variance Trade-Off ». Proceedings of the National Academy of Sciences, vol. 116, no 32, août 2019, p. 15849‑54. DOI.org (Crossref), doi:10.1073/pnas.1903070116.
>
>    [7] Lu, Zhou, et al. « The Expressive Power of Neural Networks: A View from the Width ». Advances in Neural Information Processing Systems, vol. 30, 2017, p. 6231‑39.
>
>    [8] Du, Simon, et al. « Gradient Descent Finds Global Minima of Deep Neural Networks ». International Conference on Machine Learning, PMLR, 2019, p. 1675‑85. proceedings.mlr.press, http://proceedings.mlr.press/v97/du19c.html.
>
>    [9] Goodfellow, Ian J., et al. « Qualitatively characterizing neural network optimization problems ». arXiv:1412.6544 [cs, stat], mai 2015. arXiv.org, http://arxiv.org/abs/1412.6544.

---

> ### Author Response · Authors · 2020-11-17
> **Response to Reviewer 2 (1/3)**
>
> Thank you very much for your thorough reading and the time spent reviewing our submission. This will hopefully help us improve and clarify the presentation and quality of our work.
>
> Below, we answer your questions and remarks:
>
> **"I think that the claim "We show that APHYNITY can perform similarly to complete physical models by augmenting incomplete ones, both in forecasting accuracy and physical parameter identification." is overstated. I compared APHYNITY for incomplete physics to the Param PDE or True PDE with complete physics in Table 1, and this claim seems true for the wave equation only."**
> While it is true that using APHYNITY on the incomplete dynamics doesn’t always fully compensate the incompleteness of physical models (we will soften this statement accordingly in the revised version of the paper), it still improves upon the incomplete parametric equations by several orders of magnitude in terms of MSE in all cases. The only case where there is still a significant gap with the full parametric equation is that of the reaction-diffusion equations which are indeed quite challenging and for which APHYNITY still misses some residual dynamics. However, it is important to keep in mind that forecasting results are expressed in log MSE so that the MSE differences are actually small in absolute value which means that the APHYNITY augmented incomplete dynamics still match ground truth ones quite closely as shown in Figure 3. Moreover, the APHYNITY augmentation scheme clearly outperforms a vanilla augmentation scheme and all other methods we have compared it to in Table 5 of the Appendix.
>
> **"I think that the related work section should acknowledge that people have been using neural networks for hybrids with physical models since the 1990s."**
> Due to space limitation, we have restricted the related work in the submission to the more recent and closest related works, and we acknowledge the vast body of previous works in the optimal control, robotics and system identification communities. In the revised version of the paper, we will add a few references to "gray-box" or "hybrid" methods, in particular [1] which you mentioned, and [2,3]. These methods also use neural networks in conjunction with physical models but are less generic and don’t address the issue of the uniqueness and optimality of the decomposition. They are comparable to the vanilla augmentation that we compare APHYNITY to in the ablation studies.
>
> [2] Psichogios, Dimitris C., and Lyle H. Ungar. "A hybrid neural network-first principles approach toprocess modeling." AIChE Journal 38.10 (1992): 1499-1511.[3] Thompson, Michael L., and Mark A. Kramer. "Modeling chemical processes using prior knowl-edge and neural networks." AIChE Journal 40.8 (1994): 1328-1340
>
> [3] Thompson, Michael L., and Mark A. Kramer. "Modeling chemical processes using prior knowl-edge and neural networks." AIChE Journal 40.8 (1994): 1328-1340

---

### Official Review · AnonReviewer3 · 2020-11-02
**Great paper about jointly identifying parameters of physical models and modeling the residual with a NN**

**Rating:** 9
**Confidence:** 3

**Review:**

I really enjoyed this paper. It's one of my favorite papers from 2020. Authors: thank you for writing it!

The paper proposes a framework for jointly fitting the parameters of a physical model (such as the parameters of an ODE or PDE) and learning a neural network to model the error or residual of this physical model. The idea is to find a physical dynamics model $F_p \in \mathcal{F}_p$ (e.g. where $\mathcal{F}_p$ is a set of PDEs with different parameter values) and a neural residual dynamics model $F_a \in \mathcal{F}_a$ (e.g. where $\mathcal{F}_a$ is a hypothesis class of neural networks) which minimize the norm of $F_a$ while constraining the composed dynamics $F = F_a + F_p$ to agree with observed data. Interestingly and importantly, the paper proves the minimum-norm decomposition of the observed dynamics into physical model dynamics and neural residual dynamics is unique, given a condition on the geometry of $F_p$. This condition is that $\mathcal{F}_p$ should be a Cheybshev set: a sufficient condition is that it is a closed convex set in a strict normed space. To me, this condition seems very mild.

The paper goes on to show that this method, termed APHYNITY, produces significant gains in predictive accuracy over purely learned methods, purely physics-driven methods, and other forms of combining physical models and learned models. The gain is most significant when the physical model is incomplete. When it is complete there may be some small advantage due to the neural residual accounting for some discretization error; regardless there is no harm, and the learned neural residual is very small.

The paper also shows that APHYNITY produces better parameter estimates for the physical model in the presence of incomplete physics. This seems important. Our physics models are always approximations and in many interesting applications (climate/atmosphere, bioengineering, mechanics of materials, etc) commonly used physics models may have an interestingly-sized gap with reality. When the identified parameters, not the predictions, are needed for some downstream task such as decision making, APHYNITY could help with better parameter ID.

I thought this paper was clear and well written. The main paper presents an easy-to-follow story, the appendices contain plenty of detail, and when while reading the main paper I wanted more detail on specific points, it was usually easy to follow links to the correct appendices. (Should be true of all papers, but often isn't).

---
Thoughts and feedback:
- It would be useful to see some simple visual demonstration of the effect of APHYNITY on parameter estimation in incomplete models. (I should note the paper is already quite long and thorough, though).
- The integral trajectory-based approach (section 3.2, 3.3, motivated in Appendix D) used to fit the parameters of both NN and physics model seems like the "right" way to do this to me. Nonetheless it would be interesting to see numerical comparison with the alternative (supervision over derivatives).
- I wonder if the integral trajectory-based approach, vs the supervision-over-derivatives approach, can be related to traditional methods for parameter ID in ODE?
- It seems to me that the authors do a good job explaining and relating to prior work on learning physical systems. However, with both the density of recent literature in this space and the decades of work combining ODE solvers and function approximators, if there are missing references (papers on similar work that the current submission does not cite), it's quite likely I would not have noticed.
- It would be interesting to know if this could help figure out *in what way* a physical model is misspecified, and guide design of an approximate physical model that better captures reality. I suspect yes, although it might simply boil down to this method having a better estimate of the residual than if one just does a least-squares fit with the physical model.

---
Typos:
- "bayesian" -> "Bayesian" (end page 2)
- "si" -> "is" (page 3)

---
I think this is a well written paper likely to be of interest to a large number of ICLR attendees, with some important novel contributions, and that the method proposed has a good chance of being widely adopted in the subfield of ML+physical simulation.

---

> ### Author Response · Authors · 2020-11-17
> **Response to Reviewer 3**
>
> Thank you for your positive comments and feedback! We are happy that you found our work to be useful.
> In the following, we give some additional elements to answer your questions and remarks:
>
>  **"It would be useful to see some simple visual demonstration of the effect of APHYNITY on parameter estimation in incomplete models."**
>  In Figure 1(b), we provide an illustration in the pendulum case: the incomplete physical model (which does not account for friction) predicts a smaller pendulum period than the true one.
> An additional illustration will be added for the reaction-diffusion equation in the updated version of the paper: it shows that the incomplete physical model alone underestimates the value of the diffusion coefficient for both components while APHYNITY corrects this bias and correctly identifies the diffusion coefficients.
>
> **"Nonetheless it would be interesting to see numerical comparison with the alternative (supervision over derivatives)."**
> We compare both approaches in Table 6 of the Appendix, where results show that the trajectory-based approach systematically outperforms the derivative supervision. We also believe as you do, that in general and not only for APHYNITY, optimizing over trajectories is more robust than directly approximating derivatives (argued in Section D of the Appendix), at least in terms of forecasting accuracy.
>
> **"I wonder if the integral trajectory-based approach, vs the supervision-over-derivatives approach, can be related to traditional methods for parameter ID in ODE?"**
> It is true that many methods for parameter identification [1, 2] use some form of dictionary learning and fit differential terms to an approximation of the derivative. Apart from the problem of using approximations of the derivative which we have discussed above and in the paper, this would suffer from the same kind of biases as the 'ParamODEs' used as baselines in the paper if they are not augmented with a residual model.
>
> **"It would be interesting to know if this could help figure out in what way a physical model is misspecified, and guide design of an approximate physical model that better captures reality. I suspect yes, although it might simply boil down to this method having a better estimate of the residual than if one just does a least-squares fit with the physical model."**
> This is a very interesting point. Indeed, in many challenging settings, e.g. in climate modeling, it can be very difficult to understand what a physical model lacks in order to fit measured observations more accurately. In our opinion, our approach can be leveraged in three ways in this regard:
> + The norm of the augmentation $||F_a||$ can serve as a measure quantifying the overall quality of a physical model. For example, in Table 1, we can see the difference between the Hamiltonian model and the ParamODE($\omega_0$) in the pendulum case.
> + The learned $F_p$ gives a more robust and precise estimation of what the family of physical models are able to capture.
> + The learned augmentation $F_a$ should capture all phenomena non modelled in $\mathcal{F}_p$. Analyzing this function could then provide valuable insights.
>
> It is an appealing perspective to imagine this method helping physicists craft better models of reality!
>
> Thanks for detecting the typos, they will be corrected in the revised version.
>
> [1] : https://www.pnas.org/content/113/15/3932.full
>
> [2] : https://www.semanticscholar.org/paper/Distilling-Free-Form-Natural-Laws-from-Experimental-Schmidt-Lipson/164157672985407454f5edfd92a1da287445445d

---

### Author Response · Authors · 2020-11-24
**General response and update**

Dear reviewers, we would like to thank you for your careful reading of our work and your detailed comments. We have really appreciated your constructive feedback that has hopefully helped to improve the quality of our submission. We have thus updated it (modifications in blue in the text), including:

- a more comprehensive related work section (R2 \& R1) with added historical references;

- additional visualizations in Figure 4 in order to illustrate the effect of APHYNITY on parameter identification (R3);

- clarifications on experimental results and model training (R2).

---

### Decision · Program_Chairs · 2021-01-07
**Final Decision**

**Decision:**

Accept (Oral)

**Comment:**

The authors propose a method for modeling dynamical systems that balances theoretically derived models, which may be grounded in domain knowledge but subject to overly strict assumptions, with neural networks that can pick up the slack. All reviewers were enthusiastic about this work, appreciating its balance of mathematical rigor and experimental assessment. One concern was that this paper follows on decades of related work, which was difficult to adequately summarize. However, changes made throughout discussion phase did address these concerns.